# An optimal estimation-based retrieval of upper atmospheric oxygen airglow and temperature from SCIAMACHY limb observations

Kang Sun[1,2], Mahdi Yousefi[1], Christopher Chan Miller[3,4,6], Kelly Chance[3], Gonzalo González Abad[3], Iouli E. Gordon[3], Xiong Liu[3], Ewan O'Sullivan[3], Christopher E. Sioris[5], and Steven C. Wofsy[6,7]

[1]Department of Civil, Structural and Environmental Engineering, University at Buffalo, Buffalo, NY, USA
[2]Research and Education in Energy, Environment and Water Institute, University at Buffalo, Buffalo, NY, USA
[3]Center for Astrophysics | Harvard & Smithsonian, Cambridge, MA, USA
[4]Climate Change Research Center, University of New South Wales, Sydney, New South Wales, Australia
[5]Environment and Climate Change Canada, Toronto, Ontario, Canada
[6]Harvard John A. Paulson School of Engineering and Applied Sciences, Harvard University, Cambridge, MA, USA
[7]Department of Earth and Planetary Sciences, Harvard University, Cambridge, MA, USA

**Correspondence:** Kang Sun (kangsun@buffalo.edu)

**Abstract.** An optimal estimation-based algorithm is developed to retrieve number density of excited oxygen ($O_2$) molecules that generate airglow emissions near 0.76 $\mu$m ($b^1\Sigma_g^+$ or $A$ band) and 1.27 $\mu$m ($a^1\Delta_g$ or $^1\Delta$ band) in the upper atmosphere. Both oxygen bands are important for the remote sensing of greenhouse gases. The algorithm is applied to the limb spectra observed by the SCanning Imaging Absorption spectroMeter for Atmospheric CHartographY (SCIAMACHY) instrument in both nominal (tangent heights below $\sim$90 km) and mesosphere-lower thermosphere (MLT) modes (tangent heights spanning 50–150 km). The number densities of emitting $O_2$ in the $a^1\Delta_g$ band are retrieved in an altitude range of 25–100 km near daily in 2010, providing a climatology of $O_2$ $a^1\Delta_g$ band airglow emission. This climatology will help disentangle airglow from backscattered light in nadir remote sensing of the $a^1\Delta_g$ band. The global monthly distributions of the vertical column density of emitting $O_2$ in $a^1\Delta_g$ state show mainly latitudinal dependence without other discernible geographical patterns. Temperature profiles are retrieved simultaneously from the spectral shapes of the $a^1\Delta_g$ band airglow emission in the nominal limb mode (valid altitude range 40–100 km) and from both $a^1\Delta_g$ and $b^1\Sigma_g^+$ band airglow emissions in the MLT mode (valid range 60–105 km). The temperature retrievals from both airglow bands are consistent internally and in agreement with independent observations from ACE-FTS and MIPAS with absolute mean bias near or below 5 K and root mean squared error (RMSE) near or below 10 K. The retrieved emitting $O_2$ number density and temperature provide a unique dataset for remote sensing of greenhouse gases and constraining the chemical and physical processes in the upper atmosphere.

## 1 Introduction

This study of upper atmospheric airglow from oxygen ($O_2$) is driven by the need to measure $O_2$ simultaneously with methane ($CH_4$) and carbon dioxide ($CO_2$) in satellite remote sensing. Methane and $CO_2$ are two of the most important anthropogenic greenhouse gases, but the spatiotemporal variations in their sources and sinks are poorly understood, leading to significant uncertainties in projections of future climate trends (Miller et al., 2007; Turner et al., 2019; Friedlingstein et al., 2020; Saunois

et al., 2020). Methane in particular has a global potential warming of about 56–105 times higher than that of $CO_2$ for a 20-year time period (Howarth, 2014). Reducing methane emissions is among the most impactful actions that can be taken to reduce the global warming, which requires identification of emission sources with a high degree of accuracy. Space-borne observations offer a powerful tool in quantifying the spatiotemporal distributions of methane and $CO_2$ and inferring their sources and sinks

due to the extensive spatial and temporal coverage of satellites (Eldering et al., 2017; Turner et al., 2018; Lorente et al., 2021). To separate the sources and sinks of methane and $CO_2$ from variations of surface pressure and specific humidity, the abundances of $CO_2$ and $CH_4$ observed by satellites are usually represented by the column-averaged dry mole fractions ($X_{CO_2}$ and $X_{CH_4}$). Simultaneous observation of $O_2$ is often necessary for deriving $X_{CO_2}$ and $X_{CH_4}$ and accounting for contamination of aerosol and clouds (Butz et al., 2011).

The two widely used $O_2$ absorption bands in greenhouse gas remote sensing are the $b^1\Sigma_g^+ \leftarrow X^3\Sigma_g^-$ band near 0.76 $\mu$m ($O_2$ $A$ band hereafter) and the $a^1\Delta_g \leftarrow X^3\Sigma_g^-$ band near 1.27 $\mu$m ($O_2$ $^1\Delta$ band hereafter). The $O_2$ $A$ band is commonly used in existing and planned space-borne $CO_2$ and methane sensors (Bovensmann et al., 1999; Crisp et al., 2017; Veefkind et al., 2012; Moore III et al., 2018; Buchwitz et al., 2013). The $O_2$ absorption feature in the $A$ band is much stronger than the shortwave-infrared $CO_2$ and $CH_4$ bands and separated by significant spectral gaps, which challenges the spectral fitting

and radiative transfer modeling. The $O_2$ $A$ band also overlaps with strong terrestrial solar-induced fluorescence that provides valuable information on plant photosynthesis but perturbs the $O_2$ absorption features and, if not properly accounted for, leads to systematically biased greenhouse gas retrievals (Frankenberg et al., 2012).

The $O_2$ $^1\Delta$ band plays an instrumental role in ground-based greenhouse gas remote sensing (Wunch et al., 2011; Fu et al., 2014; Frey et al., 2019) and is adopted by the MicroCarb (Bertaux et al., 2020) and MethaneSAT (Staebell et al., 2021) satel-

40 lite missions. Both $O_2$ $A$ and $^1\Delta$ bands feature upper-atmospheric airglow emissions (Zarboo et al., 2018) that overlap with the backscattered solar light containing $O_2$ absorption signals. The $A$ band airglow, which peaks in the mesopause region, has previously been neglected in nadir remote sensing due to its low intensity (Sioris, 2003), but there is a lack of quantitative assessment of how the $A$ band airglow impacts greenhouse gas retrieval. The $^1\Delta$ band airglow emitted from the upper stratosphere and mesosphere is much stronger due to photochemically generated $O_2$ in $a^1\Delta_g$ state from ozone photolysis and

45 was the main reason of choosing the $A$ band over the $^1\Delta$ band by space-borne sensors (Kuang et al., 2002). However, recent advances in modeling the $O_2$ $^1\Delta$ band airglow spectra enable disentangling the airglow from backscattered light (Sun et al., 2018a; Bertaux et al., 2020) and open up the opportunity for space-borne nadir remote sensing of the $O_2$ $^1\Delta$ band.

In order to account for the airglow contributions in the nadir-observed $O_2$ spectra, it is crucial to have an accurate understanding of the spatial, temporal, and spectral distribution of airglow emissions. Nevertheless, observation-based studies of $O_2$

$A$ and $^1\Delta$ band airglow are rare and generally lack the information needed for nadir space-borne remote sensing of greenhouse gases. Zarboo et al. (2018) retrieved volume emission rates (VER) for the $O_2$ $A$ and $^1\Delta$ band airglow over the vertical range of 50–150 km using the mesosphere and lower thermosphere (MLT) limb observation mode of the SCanning Imaging Absorption spectroMeter for Atmospheric CHartographY (SCIAMACHY) instrument. These MLT observations could not capture a significant portion of the $O_2$ $^1\Delta$ band airglow, which peaks near the stratopause, and the linear inversion applied by Zarboo et al.

(2018) is subject to systematic biases in the retrieved airglow VER below 90 km for the $O_2$ $A$ band and below 60 km for the

$O_2$ $^1\Delta$ band due to $O_2$ self-absorption (Sun et al., 2018a). Li et al. (2020) conducted high vertical and along-track resolution retrieval of the $O_2$ $^1\Delta$ band airglow for the purpose of retrieving mesospheric ozone using the OSIRIS IR imager limb radiance, which could not reveal the necessary horizontal distributions and spectral shape of airglow. Sun et al. (2018a) and Bertaux et al. (2020) fit SCIAMACHY limb spectra in the $O_2$ $^1\Delta$ band with onion-peeling algorithms and retrieved airglow from selected orbits, although the spectral variation of airglow emission lines due to temperature was not fully incorporated.

The distribution of temperature in the atmospheric regions where airglow is emitted is important to reconstruct the emission spectra. The spectral band shape has been used to retrieve upper atmospheric temperature using $CO_2$ (Boone et al., 2005; Marshall et al., 2011; García-Comas et al., 2014), $O_2$ $A$ band absorption (Nowlan et al., 2007), and the $O_2$ $A$ band airglow (Sheese et al., 2010; Yang et al., 2021). Using the $O_2$ $^1\Delta$ band airglow spectra, Sun et al. (2018a) was able to retrieve mesospheric temperature above 60 km, limited by excessive uncertainties below it. The Mass Spectrometer Incoherent Scatter (MSIS) model provides temperature estimates throughout the atmosphere and is often used when observations are unavailable (Picone et al., 2002).

To this end, we develop an optimal estimation-based algorithm to retrieve the airglow emissions and upper atmospheric temperature from limb-viewed radiance spectra. Compared with the onion-peeling algorithm in Sun et al. (2018a) that fits limb spectra from high to low tangent heights progressively, this algorithm combines spectra over a range of tangent heights and simultaneously retrieves vertical profiles of local emissions and temperature in a consistent manner. The use of Bayesian inversion enables incorporation of *a priori* knowledge, balancing of measurement error and prior error, and detailed posterior error analysis including the averaging kernel matrix and degrees of freedom for signal (DOFS) (Rodgers, 2000; Brasseur and Jacob, 2017). The airglow emission spectra are simulated based on the spectral model for the $O_2$ $^1\Delta$ band proposed by Sun et al. (2018a), which we demonstrate can be extended to the $O_2$ $A$ band with simple generalization. We compare the retrieved temperatures in the upper stratosphere, mesosphere, and lower thermosphere with independent measurements. The algorithm is applied to one year of SCIAMACHY of limb observations, including the MLT mode, to construct a climatology of $O_2$ airglow and upper atmospheric temperature at 10:00 local solar time.

## 2 Data

The SCIAMACHY limb spectra and MSIS model outputs are necessary for the retrieval algorithm. The retrieved temperature profiles from SCIAMACHY are compared with observations from Atmospheric Chemistry Experiment-Fourier-transform spectroscopy (ACE-FTS) and Michelson Interferometer for Passive Atmospheric Sounding (MIPAS) instruments.

### 2.1 SCIAMACHY

The SCIAMACHY instrument is an eight-channel grating spectrometer that measures radiation that is backscattered, reflected, transmitted, or emitted by the Earth's atmosphere and surface in limb and nadir geometries from 240 to 2380 nm. The instrument was launched on board the Envisat satellite which was operational on a sun synchronous orbit with an equator crossing time in the descending node of 10:00 local solar time from March 2002 until April 2012. In this study, only limb scattering

measurements in spectral channel 4 (597–789 nm) and 6 (990–1750 nm) were used. In nominal limb mode, SCIAMACHY observed the atmosphere from the surface up to 93 km in 2010. It also observed the MLT region covering 50–150 km altitudes in the limb view over two days in each month from July 2008 until April 2012. We conduct spectral and radiometric calibrations of SCIAMACHY spectra using version 3.2.6 of the SciaL1C command-line tool software package. We then use a customized algorithm to convert the Level 1 data to NetCDF4 file format. In contrast to previous studies (Bender et al., 2017; Sun et al., 2018a; Zarboo et al., 2018; Bertaux et al., 2020) where SCIAMACHY spectra were averaged across-track, we keep the eight across-track positions separate to enhance the horizontal space resolution. Limb observations over different altitudes at the same across-track position (usually there are 15 observing altitudes per across-track position for both nominal and MLT modes) are grouped as a single vertical "sounding" in order to retrieve atmospheric profiles. The latitude and longitude of the tangent point near 50 km is used to represent the location of the entire vertical sounding, which gives a location ambiguity of less than ∼20 km. The average vertical distance between adjacent tangent heights within the same across-track position is about 6.6 km for both nominal and MLT modes. We have noticed that the eight vertical soundings across-track can always be grouped into four pairs. The soundings within each pair are close in latitude and longitude but zigzag in tangent heights. Therefore, the eight vertical soundings can be combined into four vertical soundings with doubled vertical sampling at 3.3 km. We retrieve profiles at native eight across-track positions and only pair the retrieved profiles into four when calculating the column number density of emitting $O_2$.

To cover the *A* band, radiance spectra in the 759 nm to 772 nm range are obtained from channel 4. In addition to airglow emissions, the observed radiance spectra contain photons from Rayleigh scattering and multiple scattering by the atmosphere and the surface. The scattering signal can be approximated by limb views at maximum tangent heights and contains $O_2$ absorption feature for the *A* band. No $O_2$ absorption is observed for the $^1\Delta$ band due to much weaker atmospheric scattering and lower $O_2$ absorption. To account for the scattered light, a background signal consisting of averaged *A* band spectra at high tangent heights (130–150 km) from the same sounding is subtracted from each limb spectrum (Zarboo et al., 2018). Before each subtraction, the background signal is scaled to match its out-of-band radiance with the out-of-band radiance of the limb spectrum to be corrected for. This step assures the out-of-band radiance centers at zero after correction. This correction assumes the spectral shape of scattered light is the same for all tangent heights and may leads to systematic errors at low tangent heights from the *A* band, where airglow emission is low, and scattering path may differ significantly from the thermosphere. Following Zarboo et al. (2018), we consider 750–759 nm and 767–780 nm to be out of band. The $O_2$ $^1\Delta$ band spectra are taken from 1240 nm to 1300 nm in channel 6 with two bad pixels removed. The multiple scattering is considered negligible in the $O_2$ $^1\Delta$ band, and only a linear background fitted using out-of-band radiance at 1210–1240 nm and 1300–1340 nm is subtracted from each limb spectrum.

## 2.2 MSIS model

The temperature, pressure and the ground-state $O_2$ number densities are sampled from the NRLMSISE-00 model, a MSIS model developed at the Naval Research Laboratory (Picone et al., 2002) using a Python package by Hirsch and Kastinen (2021). The MSIS temperature profile is used as *a priori* values in the optimal estimation. The pressure profile is fixed in

the algorithm, whereas the relative changes of $O_2$ number densities from MSIS values are retrieved, which affects the $O_2$ self-absorption.

## 2.3 ACE-FTS

The ACE-FTS instrument was launched aboard the Canadian SCISAT-1 satellite in August 2003 and is still active. It operates from 750 to 4,400 cm$^{-1}$ (2.2 to 13.3 $\mu$m) at a high spectral resolution (0.02 cm$^{-1}$). A sequence of atmospheric transmission spectra in the limb geometry is recorded using the Sun as an infrared source during sunrise and sunset (i.e., solar occultation). The temperature and pressure retrievals from the ACE-FTS was performed at 15–125 km by fitting a set of spectral windows containing $CO_2$ lines with the inverse of temperature as a free parameter (Boone et al., 2020). The temperature retrieval process

was done by dividing the atmosphere into two altitude regimes with a crossover at about 50 km. The temperature retrieval was performed assuming a fixed $CO_2$ mixing ratio up to about 50 km while above it an empirical function was used to describe $CO_2$ mixing ratio in order to force the results to exhibit smooth behavior. We use the temperature profiles from version 4.1 of ACE-FTS Level 2 data for the comparison of temperature retrieved from SCIAMACHY airglow. The ACE-FTS temperature profiles have been extensively used in the validation of profiles from other instruments such as SOFIE (Marshall et al., 2011),

MIPAS (García-Comas et al., 2014), and OSIRIS (Sheese et al., 2012).

## 2.4 MIPAS

The MIPAS instrument was a Fourier transform spectrometer for the detection of limb emission spectra in the middle and upper atmosphere on-board the Envisat satellite since March 2002. It observed a wide spectral interval throughout the mid-infrared with a high spectral resolution of 0.025 cm$^{-1}$. Operating in a wavelength range from 4.15 $\mu$m to 14.6 $\mu$m, MIPAS detected and

spectrally resolved a large number of emission features of atmospheric minor constituents playing a major role in atmospheric chemistry. The retrieval of temperature is done from measurements of the $CO_2$ atmospheric radiance at 15 $\mu$m for each MIPAS single limb scan. Because MIPAS was on the same platform as SCIAMACHY, there were abundant collocated observations, enabling monthly and zonally resolved temperature intercomparison. The measurement modes of MIPAS used in this study include the nominal measurement mode with an altitude coverage of roughly 6–70 km, the middle atmosphere (MA) mode

covering 18–102 km, the upper atmospheric (UA) mode covering 42–172 km, and the noctilucent cloud (NLC) mode covering 39–102 km. The nominal measurement mode makes up the bulk of MIPAS measurements, whereas the MA and UA modes were available every at least 10 days, and the NLC mode only happened on a few days in 2010. We use the nominal temperature profiles from version 8 of MIPAS Level 2 data retrieved by ESA (Dinelli et al., 2021). Version 8 data from the other modes are obtained through the Institute of Meteorology and Climate Research in cooperation with the Instituto de Astrofísica de

Andalucía (IMK/IAA) retrieval algorithm (García-Comas et al., 2012; Kiefer et al., 2021). The typical total errors are 0.5–2 K below 70 km and 2–7 K above (for MA, UA, and NLC modes). The typical vertical resolutions in the comparison range of this study are 3–7 km.

## 3 Methodology

We assume the atmosphere as homogeneous layers and calculate airglow emission and $O_2$ self-absorption using HITRAN (Rothman, 2021). The local emission and absorption from these layers are calculated and integrated along the line-of-sight to simulate the limb-viewed radiance. A Bayesian inversion is applied to retrieve airglow emission and temperature profiles by minimizing the difference between simulated and observed limb radiance spectra with *a priori* regularization.

### 3.1 Airglow emission from a single layer

An atmospheric layer bounded by two tangent heights of SCIAMACHY limb observations is the basic spatial resolving unit of this study. We create an additional layer above the outermost tangent height by assuming a layer thickness equal to the average difference between adjacent tangent heights. The atmospheric properties are assumed to be uniform within a layer, and the layer height is represented by its middle altitude when sampling meteorological parameters from the MSIS model and comparing with other observations. The excited $O_2$ molecules that give rise to $A$ and $^1\Delta$ band airglow are denoted as $O_2(b^1\Sigma_g^+)$ and $O_2(a^1\Delta_g)$, respectively, and generalized as $O_2^*$. Their number densities ($[O_2^*]$) within the layer are proportional to the VER via the band-integrated Einstein A coefficients $a$:

$$\text{VER} = [O_2^*]a. \tag{1}$$

Here VER is measured by photons cm$^{-3}$ s$^{-1}$, $[O_2^*]$ is measured by molecules cm$^{-3}$, and Einstein A coefficient is $2.27 \times 10^{-4}$ s$^{-1}$ for the $^1\Delta$ band (Sun et al., 2018a) and $0.08693$ s$^{-1}$ for the $A$ band (Long et al., 2010). It is worth pointing out that $[O_2^*]$ is the number density that directly contributes to the airglow at the wavelengths of the $O_2$ $A$ band or $^1\Delta$ band and does not necessarily include all excited $O_2$ in the $b^1\Sigma_g^+$ or $a^1\Delta_g$ state. For example, the emission channel $b^1\Sigma_g^+$–$a^1\Delta_g$, which corresponds to the so-called Noxon band at 1.91 $\mu$m (Noxon, 1961), is ignored. Nevertheless, this does not change the derived parameters.

The absorption and emission spectra in a layer can be resolved by either wavelength ($\lambda$) or wavenumber ($\nu = 1/\lambda$). We will use the subscript $\lambda$ to denote a parameter resolved by wavelength and subscript $\nu$ to denote a parameter resolved by wavenumber. We calculate the monochromatic absorption cross section of $O_2$ using the Python library HITRAN Application Programming Interface (HAPI) (Kochanov et al., 2016) at grid space of 0.0002 nm for the $O_2$ $A$ band and 0.001 nm for the $^1\Delta$ band assuming Voigt lineshape profile. Note the monochromatic absorption cross section in cm$^2$ molecule$^{-1}$ is the same at wavelength and wavenumber coordinates, i.e., $\sigma_\lambda = \sigma_\nu$, so we may simplify it as $\sigma$. We have tested both HITRAN2016 (Gordon et al., 2017) and HITRAN2020 (Gordon et al., 2022) spectroscopic databases and provide results using HITRAN2020 unless otherwise noted. The Jacobian of $\sigma$ with respect to temperature $T$, $\partial\sigma/\partial T$, is calculated by finite difference of 0.01 K using HAPI.

The monochromatic VER (i.e., emissivity) for airglow in wavenumber space is denoted as $\varepsilon_\nu$ with a unit of photons cm$^{-3}$ s$^{-1}$ (cm$^{-1}$)$^{-1}$. Integrating $\varepsilon_\nu$ across the band should recover the VER:

$$\text{VER} = \int_\nu \varepsilon_\nu \, \mathrm{d}\nu. \tag{2}$$

According to the spectroscopic airglow model described in Sun et al. (2018a) and Bertaux et al. (2020), the $O_2$ airglow emission spectrum $\varepsilon_\nu$ is related to its absorption spectrum $\sigma$ by

$$\varepsilon_\nu = \frac{c_0 \sigma \nu^2}{\exp(c_2\nu/T) - 1},$$ (3)

where $c_0$ is a scaling constant, and $c_2$ is the second radiation constant in Planck's law with a value of $1.4387769$ cm K. Combining Eq. 1, 2, and 3, we can solve for $c_0$:

$$c_0 = \frac{[O_2^*]a}{\displaystyle\int_\nu \frac{\sigma \nu^2}{\exp(c_2\nu/T) - 1} \, d\nu}.$$ (4)

Plugging in Eq. 4 into Eq. 3:

$$\varepsilon_\nu = [O_2^*]a\frac{t_\nu}{\displaystyle\int_\nu t_\nu \, d\nu},$$ (5)

where

$$t_\nu = \frac{\sigma \nu^2}{\exp(c_2\nu/T) - 1}$$ (6)

is an intermediate function to facilitate Jacobian derivation in the following step.

The SCIAMACHY retrieval works in wavelength space, so it is more convenient to define airglow emissivity as a function of wavelength (i.e., using $\varepsilon_\lambda$ instead of $\varepsilon_\nu$). The unit of $\varepsilon_\lambda$ is photons $cm^{-3}$ $s^{-1}$ $nm^{-1}$. Integrating $\varepsilon_\lambda$ across the band in wavelength space should equally give the VER:

$$\text{VER} = \int_\lambda \varepsilon_\lambda \, d\lambda.$$ (7)

$\varepsilon_\lambda$ and $\varepsilon_\nu$ are hence related via

$$\varepsilon_\lambda = \varepsilon_\nu \frac{\nu}{\lambda},$$ (8)

and similar relationships apply for the Jacobians of $\varepsilon_\lambda$ and $\varepsilon_\nu$. The Jacobian of airglow emissivity with respect to $[O_2^*]$ is

$$\begin{aligned}
\frac{\partial \varepsilon_\lambda}{\partial [O_2^*]} &= \frac{\partial \varepsilon_\nu}{\partial [O_2^*]} \frac{\nu}{\lambda} \\
&= \frac{a\nu t_\nu}{\lambda \displaystyle\int_\nu t_\nu \, d\nu}.
\end{aligned}$$ (9)

The Jacobian of $\varepsilon_\lambda$ with respect to temperature is more complicated and can be derived by differentiating Eq. 5:

$$\frac{\partial \varepsilon_\lambda}{\partial T} = \frac{\partial \varepsilon_\nu}{\partial T} \frac{\nu}{\lambda}$$

$$= [O_2^*]a \frac{\dfrac{\partial t_\nu}{\partial T} \displaystyle\int_\nu t_\nu \, d\nu - t_\nu \displaystyle\int_\nu \dfrac{\partial t_\nu}{\partial T} \, d\nu}{\left(\displaystyle\int_\nu t_\nu \, d\nu\right)^2} \frac{\nu}{\lambda}. \tag{10}$$

Here we leverage the fact that

$$\frac{\partial \displaystyle\int_\nu t_\nu \, d\nu}{\partial T} = \int_\nu \frac{\partial t_\nu}{\partial T} \, d\nu. \tag{11}$$

Equation 10 is completed by plugging in

$$\frac{\partial t_\nu}{\partial T} = \frac{\partial \sigma}{\partial T} \frac{\nu^2}{\exp(c_2\nu/T) - 1} + \sigma \frac{c_2\nu^3 \exp(c_2\nu/T)}{\left(T\exp(c_2\nu/T) - 1\right)^2}, \tag{12}$$

which is the derivative of Eq. 6 with respect to $T$.

### 3.2 Forward model for limb-viewed airglow spectra

Assuming we include $N$ limb observations in a vertical sounding, limited by an altitude range, the number of atmospheric layers will also be $N$. We use index $i$ to denote each limb view in a collection of limb observations, and hence $i$ ranges from 1 to $N$. Figure 1 illustrates the line of sight at tangent layer $i = 2$ when there are $N = 7$ layers (and equivalently, tangent heights) under consideration. We use index $j$ to denote layers penetrated by the line of sight of limb view $i$, and $j$ ranges from $i$ to $N$. The line of sight slices through those layers twice under the assumption of homogeneous layers. To uniquely identify each segment of the line of sight, another index $l$ is introduced to denote those segments, from near to far, as shown in Fig. 1. Namely, a segment is a slant path of the line of sight through a layer. The segment index $l$ ranges from 1 to $2N - i$. A layer $j$ can be uniquely identified for any segment $l$:

$$j = \left| l - N + \frac{i-1}{2} \right| + \frac{i+1}{2}. \tag{13}$$

However, a given layer $j$ will correspond to two segments (i.e., each layer is penetrated twice):

$$l \in \{N + j - i, N - j + 1\}. \tag{14}$$

We denote the native-resolution radiance spectrum at limb view $i$ as $r_{\lambda,i}$, where the subscript $\lambda$ implies it is resolved in wavelength space. Its unit is photons cm$^{-2}$ s$^{-1}$ nm$^{-1}$ sr$^{-1}$. To model this radiance, we need to sum emissions of all segments

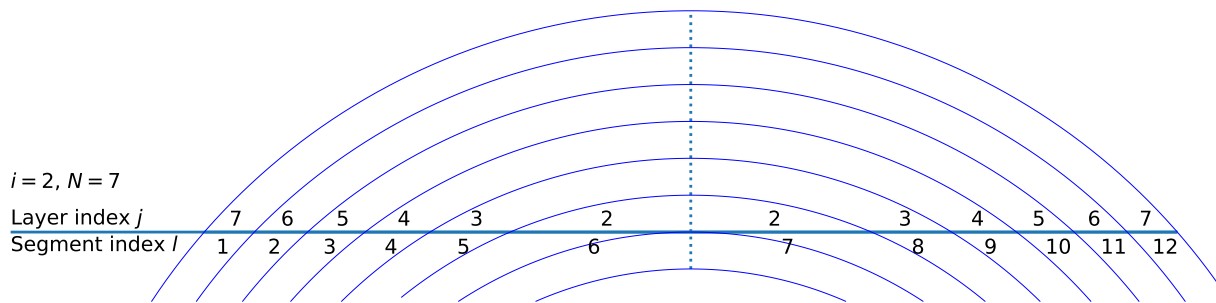

**Figure 1.** Illustration of the airglow limb-viewing line of sight. The observer is located at the left end of the horizontal line. The layer thicknesses are greatly exaggerated relative to the Earth's radius for visualization. The dashed vertical line indicates the tangent point and the thickness of each layer.

along the line of sight while considering absorption by downstream ground-state $O_2$ molecules:

$$r_{\lambda,i} = \sum_{l=1}^{2N-i} \left( \frac{L_{ij}\varepsilon_{\lambda,j}}{4\pi} \exp\left( -\tilde{\tau}_{\lambda,ij} - \sum_{l'=1}^{l-1} \tau_{\lambda,ij'} \right) \right). \tag{15}$$

Here the outermost summation loops from the nearest ($l = 1$) to the farthest ($l = 2N - i$) segment, and each segment $l$ maps uniquely to a layer $j$ through Eq. 13. $L_{ij}$ is the path length of the tangent view $i$ through layer $j$ in unit of cm. $L_{ij}\varepsilon_{\lambda,j}/(4\pi)$ is the unattenuated radiance emitted from such a segment. If there were no absorption, the exponential term in Eq. 15 will be absent, and the observed limb radiance is simply the sum of radiances of all segments.

The exponential term in Eq. 15 represents the $O_2$ self-absorption between the emitting segment $l$ and the observer. The summation within the exponential term gives the total absorption optical depth of the downstream segments ranging from $l' = l - 1$, the segment immediately downstream, to $l' = 1$, the segment closest to the observer. Each downstream segment $l'$ again maps uniquely to a layer $j'$, whose optical depth is given by

$$\tau_{\lambda,ij'} = [O_2]_{j'}\sigma_{\lambda,j'}L_{ij'}. \tag{16}$$

Here $[O_2]_{j'}$ is the $O_2$ number density (in ground-state, not to be confused with the emitting $O_2$, or $O_2^*$) in layer $j'$, and $\sigma_{\lambda,j'}$ is the monochromatic absorption cross section of $O_2$ at layer $j'$ (the subscript $\lambda$ is added back to emphasize that $\sigma$ is resolved in wavelength space).

The $\tilde{\tau}_{\lambda,ij}$ term in Eq. 15 represents the self-absorption happening within the emitting segment. It has been neglected in the onion-peeling algorithm in Sun et al. (2018a), leading to large errors in spectral fitting at lower tangent heights where self-absorption is significant. We treat it here as an "effective" optical depth as if all emitters are concentrated at one end of the

segment. It can be proven that

$$\tilde{\tau}_{\lambda,ij} = -\ln\left(\frac{1 - \exp\left(-[O_2]_j \sigma_{\lambda,j} L_{ij}\right)}{[O_2]_j \sigma_{\lambda,j} L_{ij}}\right). \tag{17}$$

Moreover, such an effective optical depth of the emitting segment approaches half of its optical depth as a transmitting-only segment (i.e., $([O_2]_j \sigma_{\lambda,j} L_{ij})/2$) when it is optically thin. The proof is given by Appendix A.

Equation 15 serves as the forward model for limb-viewed radiance, and the Jacobians with respect to parameters to be retrieved (i.e., the state vector elements) can be obtained by finite difference. To reduce computing cost, we derive the analytical Jacobians when possible. The derivations of analytical Jacobians with respect to temperature, emitting $O_2$ number density, and the relative change of ground-state $O_2$ number density are provided in Appendix B. The analytical Jacobians are also validated with finite difference. In addition, we include a squeeze factor of the SCIAMACHY instrument line shape (ILS) and a wavelength shift in the state vector. Before each inversion iteration, the radiance and Jacobians at all tangent heights are concatenated, convolved with the ILS, and sampled at the shifted SCIAMACHY wavelength grid. The Jacobians with respect to all state vector elements are then assembled into the full Jacobian matrix to be used in the inversion.

### 3.3 Optimal estimation-based inversion

After background subtraction, the limb-viewed radiance from tangent heights within the retrieval range is concatenated as the measurement vector $\mathbf{r}$. The forward model described in the previous section maps the state vector $\mathbf{x}$ into the measurement space and is denoted as $\mathbf{f}(\mathbf{x})$:

$$\mathbf{r} = \mathbf{f}(\mathbf{x}) + \mathbf{e}, \tag{18}$$

where $\mathbf{e}$ is the measurement error representing SCIAMACHY detector noise and any other inadequacy of the retrieval system to reproduce the limb-observed radiance. We denote the error covariance matrix of $\mathbf{e}$ as $\mathbf{S_r}$ and assume no correlation between individual measurements. The diagonal elements of $\mathbf{S_r}$ are calculated as the sum of a scaled radiance and the square of readout noise. The scaled radiance term approximates the detector shot noise and model-data discrepancy, whose variances are assumed to scale linearly with radiance. The scaling factors applied to the radiance are tuned to balance the model-data discrepancy and retrieval-prior discrepancy. Their values are chosen to be $5 \times 10^8$ for $O_2$ $^1\Delta$ band and $1 \times 10^7$ for the $O_2$ $A$ band. The readout noise is approximated by the standard deviation of the out-of-band radiance for each limb-viewed spectrum.

The retrieval algorithm minimizes the cost function

$$\chi^2 = \left(\mathbf{r} - \mathbf{f}(\mathbf{x})\right)^T \mathbf{S_r}^{-1} \left(\mathbf{r} - \mathbf{f}(\mathbf{x})\right) + \left(\mathbf{x} - \mathbf{x_a}\right)^T \mathbf{S_a}^{-1} \left(\mathbf{x} - \mathbf{x_a}\right), \tag{19}$$

where $\mathbf{x_a}$ is the prior vector, and $\mathbf{S_a}$ is the prior error covariance matrix. A smaller prior error leads to heavier influence by the prior values and vice versa. For each retrieval, we first conduct a linear inversion of the spectra and use the vertical mean value of the inverted $[O_2^*]$ profile as the prior values for the $[O_2^*]$ profile. Note that the linear inversion ignores self-absorption and leads to inaccurate profile shape (Sun et al., 2018a), but the purpose here is only to get a very rough estimate of $[O_2^*]$.

The corresponding prior error is set to be 100 times the prior, a constant value for all altitudes. This effectively gives no prior constraint to the $[O_2^*]$ profile and assures its information all comes from observations through near-unity DOFS of retrieved $[O_2^*]$. The *a priori* temperature profile is sampled from the MSIS model. Retrieval of temperature becomes difficult for the $O_2$ $^1\Delta$ band in the stratosphere because of strong self-absorption and interference from upper layers. To stabilize the temperature retrieval, we apply a tighter prior error of 10 K below 50 km and relax to 30 K above 50 km. A logistic function with a steepness scale of 2.5 km is applied to smooth the transition at 50 km. Above 90 km, which is only observed in the MLT mode, the temperature prior error is further relaxed to 60 K to account for the potentially large observation-prior difference in the thermosphere. In the state vector, we also include changes of the $O_2$ profile relative to the *a priori* from MSIS, which is equivalent to retrieving the natural log of $[O_2]$. The prior values are set to be zero, i.e., no change to $[O_2]$ from MSIS, and the prior errors are set at 50%. The $O_2$ number density in the fitting is not constrained with temperature and pressure through ideal gas law or hydrostatic balance. This lends some freedom for the $O_2$ number density to deviate from the MSIS model and slightly enhances the goodness of fit in layers with large self-absorption. The DOFS of the retrieved $\ln([O_2])$ also serves as a good qualitative indicator for the emergence of $O_2$ self-absorption when going from high to low altitudes. The prior errors of each individual profile ($[O_2^*]$, temperature, and $\ln([O_2])$) are correlated internally with a length scale of one scale height.

The forward model is nonlinear, so the cost function has to be minimized iteratively. We adopt the Levenberg-Marquardt modification of the Gauss-Newton method. On each iteration $i$, we solve for the state vector update $d\mathbf{x_{i+1}}$:

$$d\mathbf{x_{i+1}} = \left((1+\gamma)\mathbf{S_a}^{-1} + \mathbf{K_i}^T\mathbf{S_r}^{-1}\mathbf{K_i}\right)^{-1} \left(\mathbf{K_i}^T\mathbf{S_r}^{-1}(\mathbf{r} - \mathbf{f}(\mathbf{x_i})) - \mathbf{S_a}^{-1}(\mathbf{x_i} - \mathbf{x_a})\right), \tag{20}$$

where $\mathbf{K_i}$ is the full Jacobian matrix (i.e., $\partial\mathbf{r}/\partial\mathbf{x}$) at iteration $i$, and the state vector is initialized by the *a priori*, i.e., $\mathbf{x_0} = \mathbf{x_a}$. $\gamma$ is the Levenberg-Marquardt parameter that helps stabilize the iteration compared with the standard Gauss-Newton method. We initialize and update the value of $\gamma$ and determine convergence following the OCO-2 retrieval algorithm (Crisp et al., 2021).

After convergence is achieved, the posterior error covariance matrix can be calculated as

$$\hat{\mathbf{S}} = \left(\mathbf{K}^T\mathbf{S_r}^{-1}\mathbf{K} + \mathbf{S_a}^{-1}\right)^{-1}, \tag{21}$$

where $\mathbf{K}$ is the Jacobian matrix in the final iteration, and the averaging kernel matrix is

$$\mathbf{A} = \hat{\mathbf{S}}\mathbf{K}^T\mathbf{S_r}^{-1}\mathbf{K}. \tag{22}$$

In this study, the DOFS of each state vector element refers to the corresponding diagonal element in the averaging kernel (Liu et al., 2010).

# 4   Results

The algorithm described in the previous section is applied to retrieve $O_2$ $^1\Delta$ band airglow from SCIAMACHY nominal limb observations and both $^1\Delta$ and $A$ band airglow from SCIAMACHY MLT mode limb observations. For nominal limb observations, the measurement data are limited to tangent heights above 25 km and below 100 km (usually the uppermost tangent

height for nominal limb scans is at $\sim 90$ km). When the solar zenith angle (SZA) is above $70°$, the lower height bound is gradually increased to 40 km as the airglow vanishes at lower altitude. The *A* band airglow retrieval is not attempted using the nominal limb observations due to limited coverage.

## 4.1 Retrieval demonstration using individual vertical soundings

Figure 2 demonstrates the spectral fitting of a vertical sounding of SCIAMACHY $O_2$ $^1\Delta$ band nominal limb spectra on 3 January 2010 at $28.0°$ N, $99.5°$ E. Ten limb observations (i.e., $N = 10$) with tangent heights ranging from 28.4 km to 87.4 km are included. Figure 2a shows the concatenated radiance spectra, and Fig 2b shows the fitting residuals. The predicted 95% confidence intervals, approximated by twice the measurement error, are also overlaid with the residuals. The measurement errors at the $^1\Delta$ band are dominated by the portion that scales with radiance, whereas readout noise that is constant across the band is relatively insignificant except at the highest two tangent heights. This highlights the importance to appropriately consider the portion of measurement errors that changes with radiance. The measurement error would be grossly underestimated if only the out-of-band variance were considered. The goodness of fitting is indicated by the $\chi^2$ value of 1.74 at convergence. Retrieval using the HITRAN2016 line list gives a slightly higher $\chi^2$ of 1.79 (result not shown), indicating a more accurate $O_2$ $^1\Delta$ band spectroscopy in HITRAN2020. Indeed, the latest edition of the database makes use of very accurate experimental work that was recently carried out in Grenoble (Konefał et al., 2020; Tran et al., 2020) and NIST (Fleurbaey et al., 2021). These measurements allowed improved spectroscopic models that efficiently decouple contributions of the magnetic dipole and much weaker but not negligible electric quadrupole transitions. The theoretical background is provided in Gordon et al. (2010) and Mishra et al. (2011).

The retrieved atmospheric profiles from the vertical sounding in Fig. 2 are shown in Fig. 3. The retrieved $O_2(a^1\Delta_g)$ number density is displayed in Fig. 3a with error bars indicating twice the posterior error or approximately the 95% confidence interval. As demonstrated in previous studies, $O_2$ $^1\Delta$ band airglow concentrates in the lower mesosphere and upper stratosphere and peaks above the stratopause. The retrieved $[O_2(a^1\Delta_g)]$ becomes increasingly uncertain down to the stratosphere because (1) the lower the layer is, the smaller the number of tangent views can detect them, (2) those limb views include the contributions by emissions from all layers above, and (3) $O_2$ self-absorption becomes significant. The nominal limb observations that stop at $\sim$90 km capture most of the upper tail of $O_2$ $^1\Delta$ band airglow, as supported by comparison to the results of the MLT mode retrieval shown in Fig. 4 and Fig. 5 that extend beyond 120 km. The DOFS for the retrieved $[O_2(a^1\Delta_g)]$ is effectively unity at all retrieval altitudes as shown in Fig. 3c. The retrieved temperature profile with posterior uncertainty is shown in Fig. 3b together with MSIS temperature that is used for prior values (green) and a collocated MIPAS temperature profile (black). The MIPAS observation is located at $28.7°$ N, $99.1°$ E with a 91-km spatial separation and a 15-min temporal separation from SCIAMACHY. In the mesosphere, the retrieved SCIAMACHY temperature deviates from prior values and matches closely with independent MIPAS observations. This lends confidence to the spectral modeling of $O_2$ $^1\Delta$ band airglow and the forward model for limb-viewed spectra. The DOFS for the retrieved temperature is close to unity above 50 km and quickly drops below it (Fig. 3c), indicating heavier prior influence for the stratospheric temperature. This is by design as it is challenging to extract observational information for the stratosphere due to $O_2$ self-absorption, whereas the prior stratospheric temperature is more

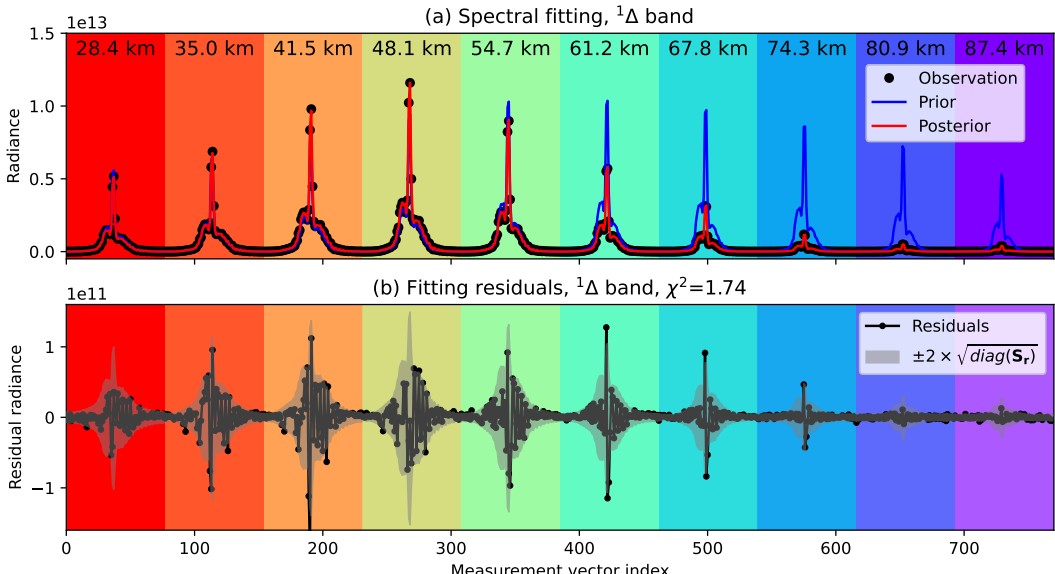

**Figure 2.** (a) SCIAMACHY limb-viewed spectra concatenated over ten tangent heights within one vertical sounding (black), simulated concatenated spectra using the prior values of the state vector (blue), and simulated concatenated spectra using the posterior values of the state vector (red). Each limb-viewed $^1\Delta$ band spectrum contains 77 data points, so the measurement vector length is 770. Altitudes of each tangent height are labeled above the corresponding spectra. The radiance unit is photons cm$^{-2}$ s$^{-1}$ nm$^{-1}$ sr$^{-1}$. (b) Residuals from fit in (a) and the 95% confidence intervals of residuals.

trustworthy. Also shown in Fig. 3c is the DOFS for the relative changes of O$_2$ number density, or equivalently $\ln([O_2])$. The influence of O$_2$ self-absorption is felt by the retrieval below 80 km and becomes significant below 60 km. Below 40 km, the
DOFS drops due to the general loss of observational constraint. The DOFS for $[O_2(a^1\Delta_g)]$ does not appear to drop because its prior constraint is much lower.

     The same O$_2$ $^1\Delta$ band retrieval is applied to an MLT mode vertical sounding on 4 January 2010 at 55.8° N, 92.0° E. The spectral fitting is shown in Fig. 4. The lowest tangent height of this particular MLT sounding is at 57 km, and the upper limit is set to below 120 km, above which the airglow is negligible. The retrieval $\chi^2$ of 0.70 (0.74 if using HITRAN2016) is
significantly lower than the nominal vertical sounding, indicating a better fitting quality. This is consistent with the fact that most challenges of the $^1\Delta$ band retrieval are in the stratosphere, which is not observed in the MLT mode.

     Figure 5 shows profiles retrieved from the spectral fit in Fig. 4. The portions below 90 km are qualitatively similar to the nominal mode retrieval in Fig. 3, although the MLT mode reveals further information at the mesopause region. There is a minor peak of $[O_2(a^1\Delta_g)]$ at near 90 km whose maximum is $\sim$20 times lower than the major peak near the stratopause (comparing
Fig. 5a and Fig. 3a). The nominal vertical soundings miss the upper part of the minor peak. Figure 5b compares retrieved MLT temperature with MSIS temperature (green) and collocated ACE-FTS temperature (black). The ACE-FTS observation is located at 55.5° N, 92.7° E with a spatial separation of 56 km and a temporal separation of 1 hour 57 min from SCIAMACHY.

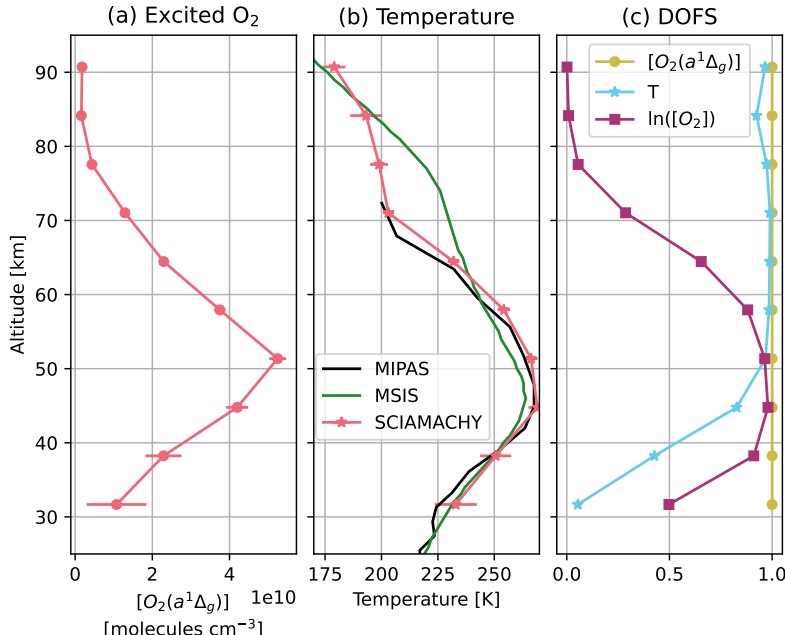

**Figure 3.** (a) Retrieved $[O_2(a^1\Delta_g)]$ profile from the spectral fitting in Fig. 2. (b) Retrieved temperature profile simultaneously with $[O_2(a^1\Delta_g)]$ (pink), collocated MSIS model temperature (green), and collocated MIPAS observation (black). Error bars indicate twice the posterior error in (a) and (b). (c) DOFS for the retrieved $[O_2(a^1\Delta_g)]$, temperature, and $\ln([O_2])$ profiles. Note that the profiles are plotted at the layer center altitude that is $\sim$3.3 km above each tangent height.

The retrieved temperature from SCIAMACHY $^1\Delta$ band airglow agrees well with ACE-FTS in the upper mesosphere but becomes more uncertain above 95 km, where the $^1\Delta$ band airglow becomes too dim to adequately reflect temperature from its

spectral shape. The behaviors of DOFS in Fig. 5c are consistent with the nominal mode retrieval where they overlap and shows the loss of observational constraint on temperature above 95 km due to fading airglow.

The O$_2$ $A$ band retrieval is applied to the same MLT mode vertical sounding and shown in Fig. 6. The upper limit of tangent height is increased to 130 km as the $A$ band airglow extends further into the thermosphere. The airglow radiance observed at the $A$ band is two orders of magnitude lower than the $^1\Delta$ band and closer to the readout noise level, which appears to dominate

the total measurement uncertainty. The residual does not show significantly larger variance at high radiance, in contrast to the $^1\Delta$ band. Overall, the $A$ band airglow spectra shapes are well captured. The $\chi^2$ value for this fit is 0.77 with little difference between HITRAN2016 and HITRAN2020.

Figure 7 shows retrieved profiles from the $A$ band spectral fitting in Fig. 6. The emitting O$_2$ number density ($[O_2(b^1\Sigma_g^+)]$) can be retrieved with high precision above $\sim 90$ km. The retrieval error quickly grows below the mesopause region (below 80

365    km) due to O$_2$ self-absorption and loss of observational constraint. The temperature profiles from both $A$ band and $^1\Delta$ band airglow are shown together with MSIS and ACE-FTS temperature profiles in Fig. 7b. Because the $A$ band airglow extends

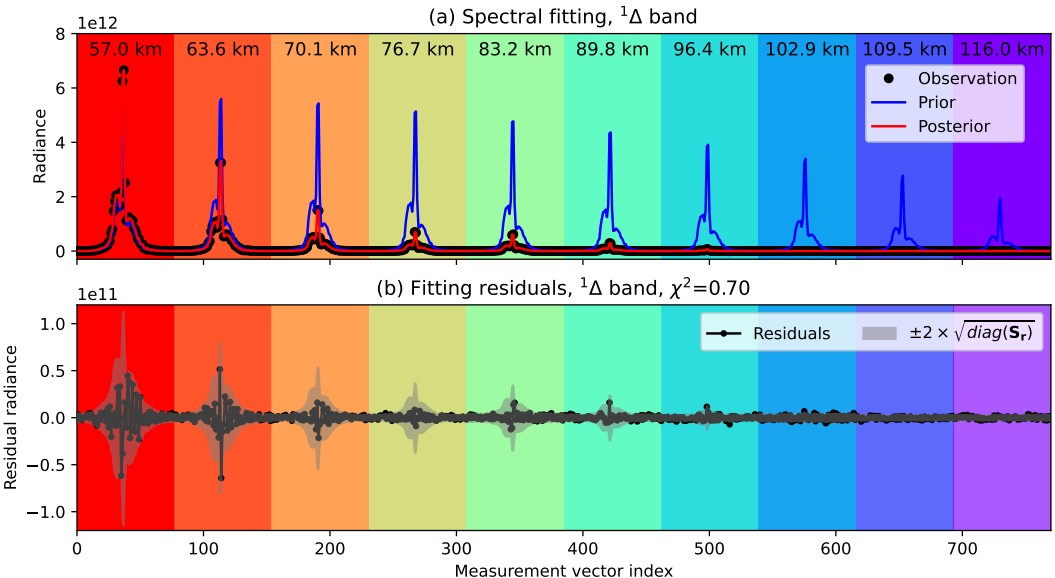

**Figure 4.** Similar to Fig. 2 but using 10 limb views in an MLT vertical sounding.

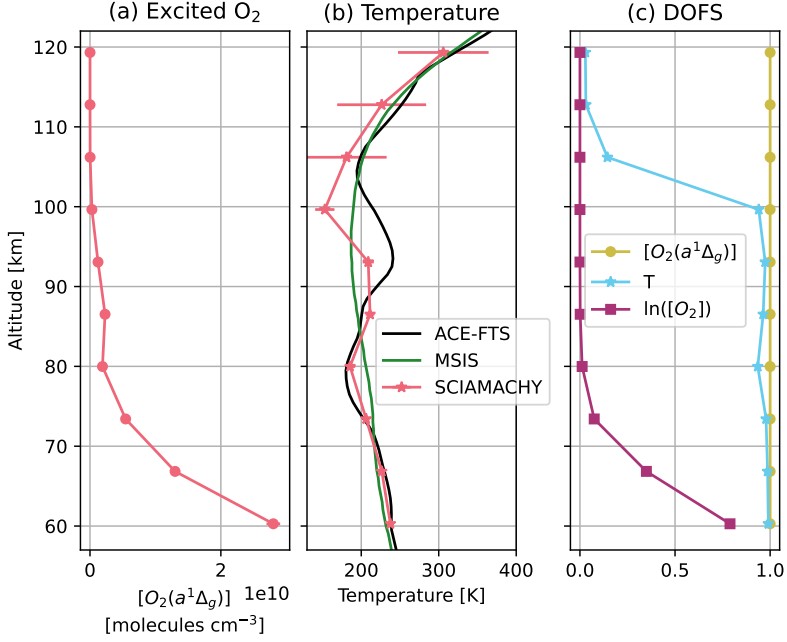

**Figure 5.** Similar to Fig. 3, but using an MLT vertical sounding.

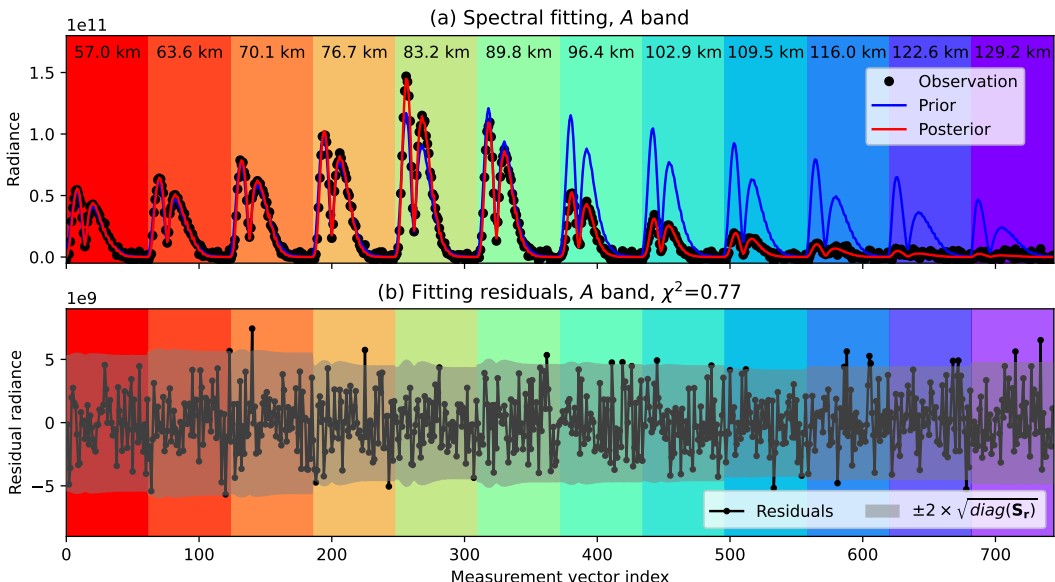

**Figure 6.** Similar to Fig. 4 but showing $O_2$ *A* band fitting using 12 limb views from the same MLT vertical sounding. Each limb-viewed *A* band spectrum contains 62 data points, so the entire measurement vector length has 744 elements.

further into the thermosphere, the *A* band-derived temperature shows less uncertainty than the $^1\Delta$ band-derived temperature above the mesopause. Below the mesopause, the $^1\Delta$ band-derived temperature shows better precision. Both SCIAMACHY-based temperature profiles capture the double mesopause feature observed by ACE-FTS and agree well between each other at

80–95 km, where both airglow are strong and less interfered. These indicate that the airglow spectral model can be reliably applied to both $O_2$ *A* and $^1\Delta$ bands.

## 4.2   Climatology of $O_2$ $^1\Delta$ band airglow and temperature retrieved from nominal limb mode observations

The $[O_2(a^1\Delta_g)]$ profiles are retrieved at eight vertical soundings across track. Numerical artifact shows up when integrating the $[O_2(a^1\Delta_g)]$ profiles vertically, as four vertical soundings are defined at staggered altitudes compared to the other four.

Therefore, we interweave the $[O_2(a^1\Delta_g)]$ profiles retrieved from the adjacent two across-track positions into one single profile and integrate the interwoven profile to obtain the column number density of $O_2(a^1\Delta_g)$. As a result, the $O_2(a^1\Delta_g)$ column number density is defined at four across-track positions instead of eight. The latitude and longitude of the paired across-track position are averaged from the ones of the original vertical soundings. Figure 8 exemplifies the column number density of $O_2(a^1\Delta_g)$ calculated from one day of retrieval on 3 January 2010 using the SCIAMACHY nominal limb observations. One

out of the 14 orbits (orbit # 41015) in Fig. 8 is highlighted to show the spatial extent of a single orbit. Some missing sounding locations are identifiable and due to failure to convergence. These missing points are generally located at high latitudes and high solar zenith angles. In these transition regions between daytime and nighttime, the horizontal variation of airglow intensity

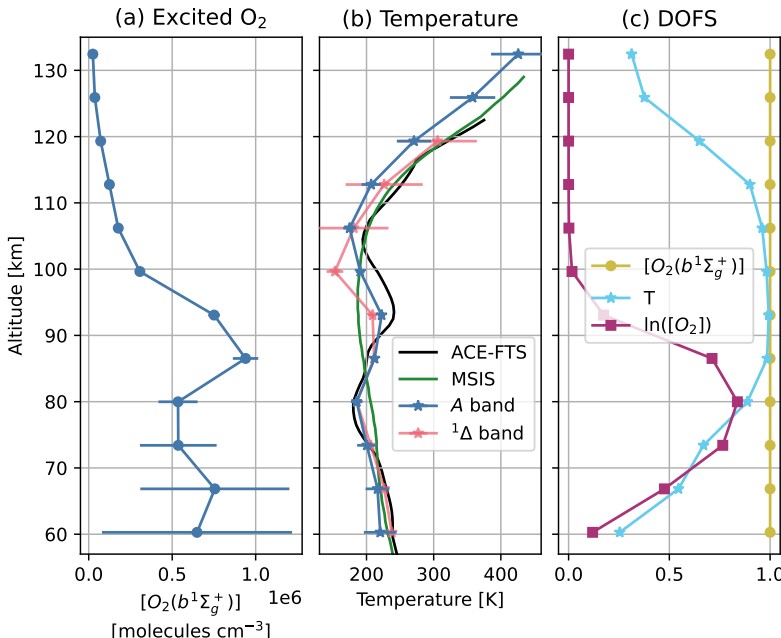

**Figure 7.** Similar to Fig. 5, but in the *A* band. The *A* band-derived emitting $O_2$ number density and temperature are shown in blue. In (b), the $^1\Delta$ band-derived temperature, the same one shown in Fig. 5b, is overlaid in pink color.

is significant, which violates the homogeneous layer assumption for the retrieval algorithm. Retrieved data are often available for part of the ascending phase of the orbit at the summer hemisphere (most valid data are in the descending phase), leading to some repeated observations at the same latitude, although at different SZAs and potentially in the nighttime. To eliminate such a latitudinal ambiguity and nighttime data, we remove the ascending portion when averaging over multiple days and limit the SZA to within $100°$. The temperature sounding locations of MIPAS on the same day are shown in Fig. 8 as small gray and black symbols. The MIPAS temperature will be used for comparison in Section 5.

We then apply the retrieval algorithm to the $O_2$ $^1\Delta$ band spectra from all SCIAMACHY nominal limb observations over the year 2010. The use of HITRAN2020 improves the goodness of fit in mid to high latitude but not in the tropical region, especially over land. This is not fully understood as HITRAN2020 clearly reduces some structural residuals through more accurate line intensities. Figure 9 illustrates the global maps of monthly averaged $O_2(a^1\Delta_g)$ column number density in 2010. The sounding locations are binned to a $3\times3°$ grid using the drop-in-the-box method (Sun et al., 2018b). Although the single-day coverage is sparse (Fig. 8), continuous coverage is achievable at monthly scale. In general, the value in each grid box is averaged from 5–10 sounding profiles. The maximum abundance of emitting $O_2$ follows the amount of solar radiation closely and generally collocates with the subsolar latitude. The zonal variation is insignificant, and no other spatial pattern of $O_2$ $^1\Delta$ band airglow can be convincingly identified. Figure 10 displays the vertical and latitudinal distribution of $O_2$ $^1\Delta$ band airglow, represented by the zonal mean of retrieved $[O_2(a^1\Delta_g)]$ profiles for each month in 2010. The seasonal shift of airglow following

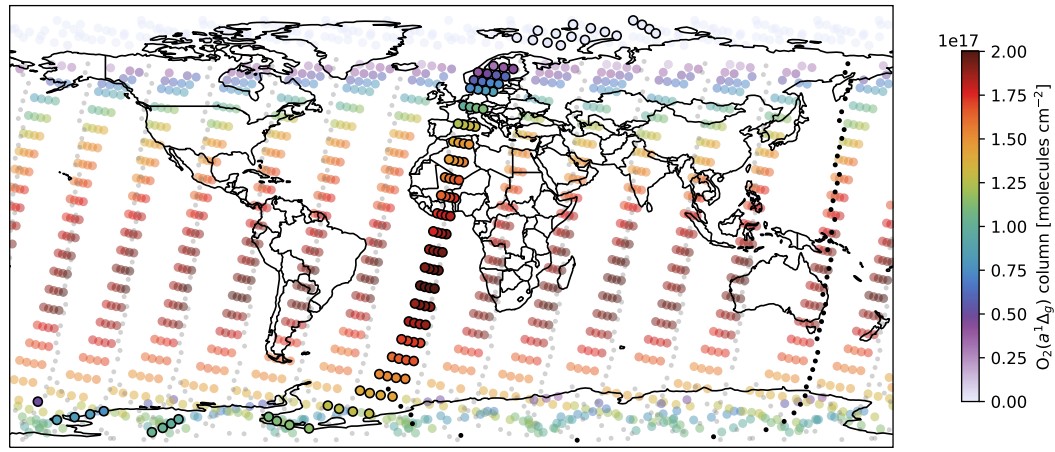

**Figure 8.** Large symbols are colored by $O_2(a^1\Delta_g)$ column number density retrieved from SCIAMACHY nominal limb orbits #41010–41023 on 3 January 2010. One SCIAMACHY orbit (# 41015) is highlighted by opaque face color and black edge color. Small, gray symbols mark the temperature sounding locations from MIPAS that shares orbits with SCIAMACHY. One MIPAS orbit (# 41023) is highlighted by black color.

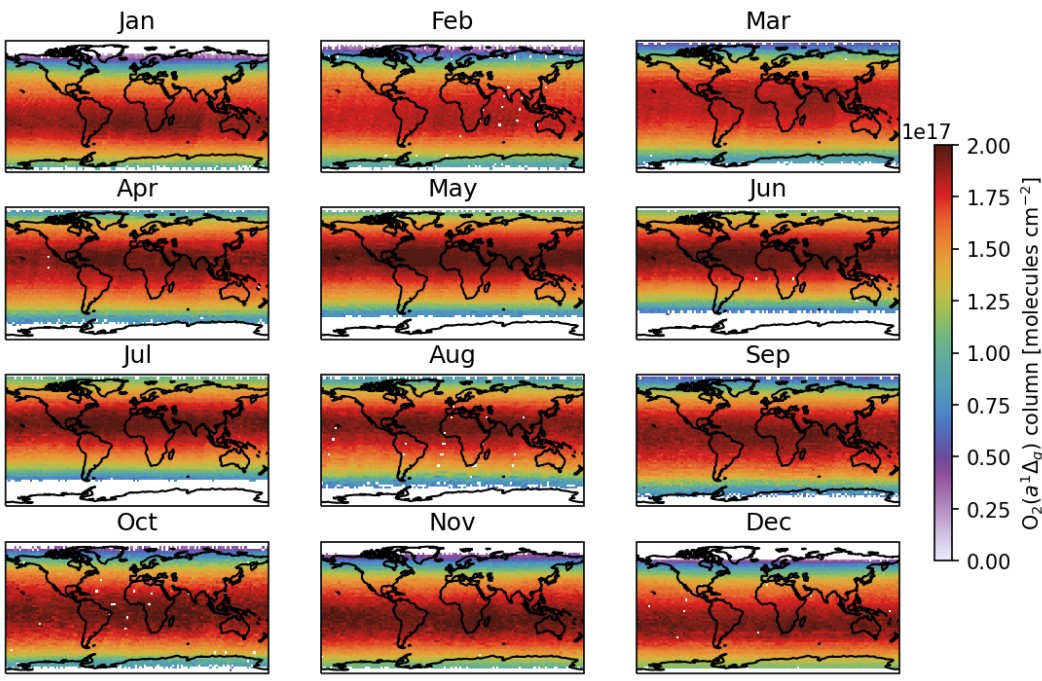

**Figure 9.** Monthly average of $O_2(a^1\Delta_g)$ column number density retrieved from SCIAMACHY nominal limb orbits.

the subsolar latitude is consistent with the spatial distribution of $O_2(a^1\Delta_g)$ column number density in Fig. 9. The vertical peak
locations of $[O_2(a^1\Delta_g)]$ at 45–50 km does not appear to change significantly over the low and mid latitudes or different time
of year. The spatial and temporal distributions of $O_2$ $^1\Delta$ band airglow are in line with previous reports using SCIAMACHY
and OSIRIS IR imager limb observations (Wiensz, 2005; Bertaux et al., 2020; Li et al., 2020) while provide a more complete
characterization.

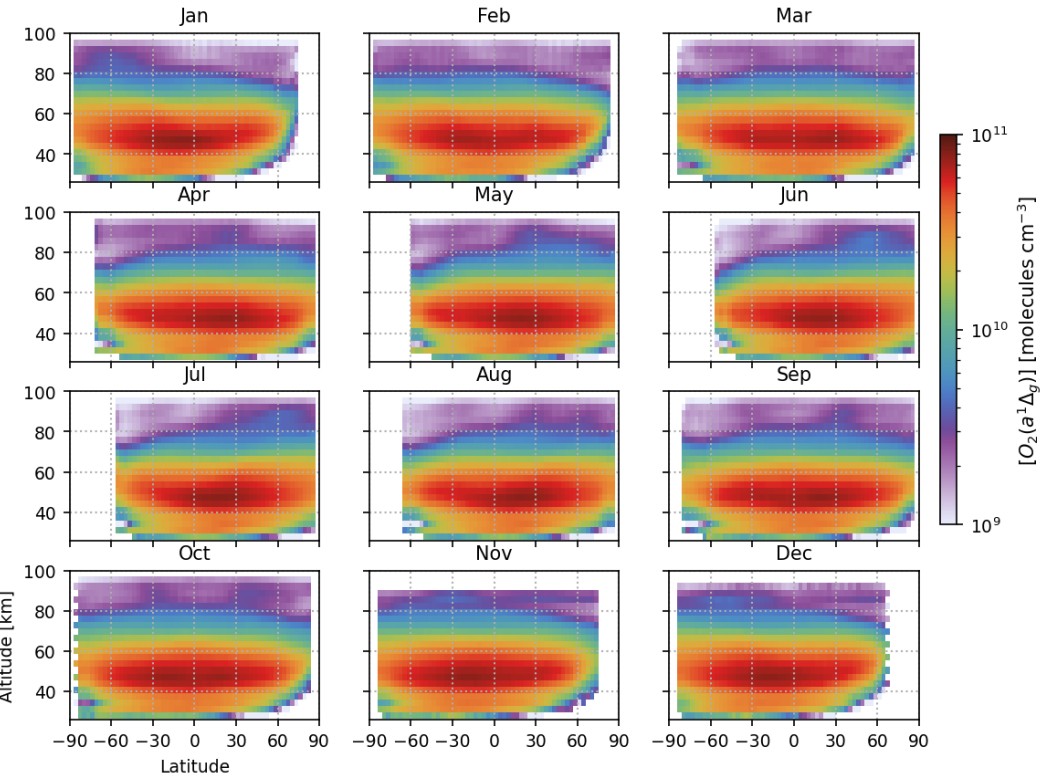

**Figure 10.** Vertical-latitudinal distribution of $[O_2(a^1\Delta_g)]$. Each data point in retrieved profiles is binned to 3D grid boxes with size of $3°\times3°\times3.2$ km and then averaged zonally.

The $O_2$ $^1\Delta$ band airglow is generated by several mechanisms (Zarboo et al., 2018; He et al., 2019; Bertaux et al., 2020), and
405 the most import mechanism of $O_2(a^1\Delta_g)$ production is the solar UV photolysis of ozone. The primary ozone layer exists in the
stratosphere, resulting from the photolysis via the the Herzberg continuum. A secondary ozone layer, generated by photolysis
via the the Schumann–Runge continuum, has been observed in the mesopause region (Smith and Marsh, 2005; Li et al., 2020).
The $O_2$ $^1\Delta$ band airglow also features two peaks. The declining ozone concentration and increasing UV with altitude give rise
to the primary peak of $[O_2(a^1\Delta_g)]$ at 45–50 km at low and mid latitudes. Figure 11 gives a closer look at the dependency
of $[O_2(a^1\Delta_g)]$ profiles on the SZA. The retrieved $[O_2(a^1\Delta_g)]$ profiles north of the subsolar latitude over January 2010 are
averaged over 2° SZA bins. The $[O_2(a^1\Delta_g)]$ profile shape shows little variation for SZA $< 50°$, whereas the primary airglow

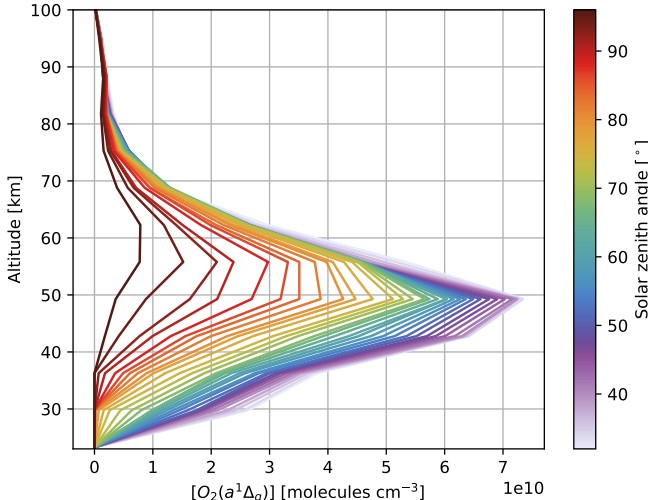

**Figure 11.** Binned profiles of $[O_2(a^1\Delta_g)]$ for SZAs between $32°$–$96°$ at $2°$ intervals during January 2010. Only data north of the subsolar latitude are used.

peak shifts to higher altitudes as SZA further increases. At the maximum observed SZA of $96°$ (left most line in Fig. 11), the primary peak locates around 60 km. This is because at large SZAs, the solar UV radiation no longer penetrates into the stratosphere and is mostly absorbed by ozone. The secondary $[O_2(a^1\Delta_g)]$ peak corresponding to the mesopause ozone layer is
vaguely identifiable at 80–100 km and does not vary much over the SZA range shown in Fig. 11.

Vertical and latitudinal distributions of temperature retrieved from $O_2$ $^1\Delta$ airglow in nominal limb mode is shown in Fig. 12 for each month in 2010. Only temperature retrievals with DOFS greater than 0.5 are included in the averaging, and only grid cells with more than 5 measurements averaged are shown, which limits the temperature profiles to above $\sim 40$ km. The retrieved temperature profiles peak at the stratopause region (40–60 km) due to ozone absorption of the solar UV radiation.
The temperature minima in the mesopause region (80–100 km) are also largely captured. Temperature in the mesopause region is determined by a combination of radiative effects, including the absorptive heating of solar UV radiation by $O_2$ and ozone (Mlynczak and Solomon, 1993) and radiative cooling of $CO_2$ infrared emission (Rodgers et al., 1992), chemical effects of exothermic reactions of odd hydrogen and odd oxygen (Mlynczak and Solomon, 1993), and the dynamics of stratosphere– mesosphere, including turbulence heating (Fritts and Vanzandt, 1993; Lübken et al., 1993) and vertical transfer of heat by
up-winds and gravity waves over the summer mesosphere. As shown in Fig 12, the mesopause temperature is cold in polar summer and relatively warm in polar winter, driven by dynamical effects. In the polar summer, stronger radiative heating from ozone and $O_2$ leads to upward air motion in the mesosphere, which in turn causes an adiabatic expansion of rising air and cools the mesopause. The upward motion that drives the cooling process is amplified by breaking upward gravity waves in the polar summer caused by the stratospheric easterly winds. An opposite process in the winter pole causes an adiabatic compression
that results in a warm mesosphere (Björn, 1984; Smith, 2004). Another interesting feature in Fig. 12 is the implausibly high

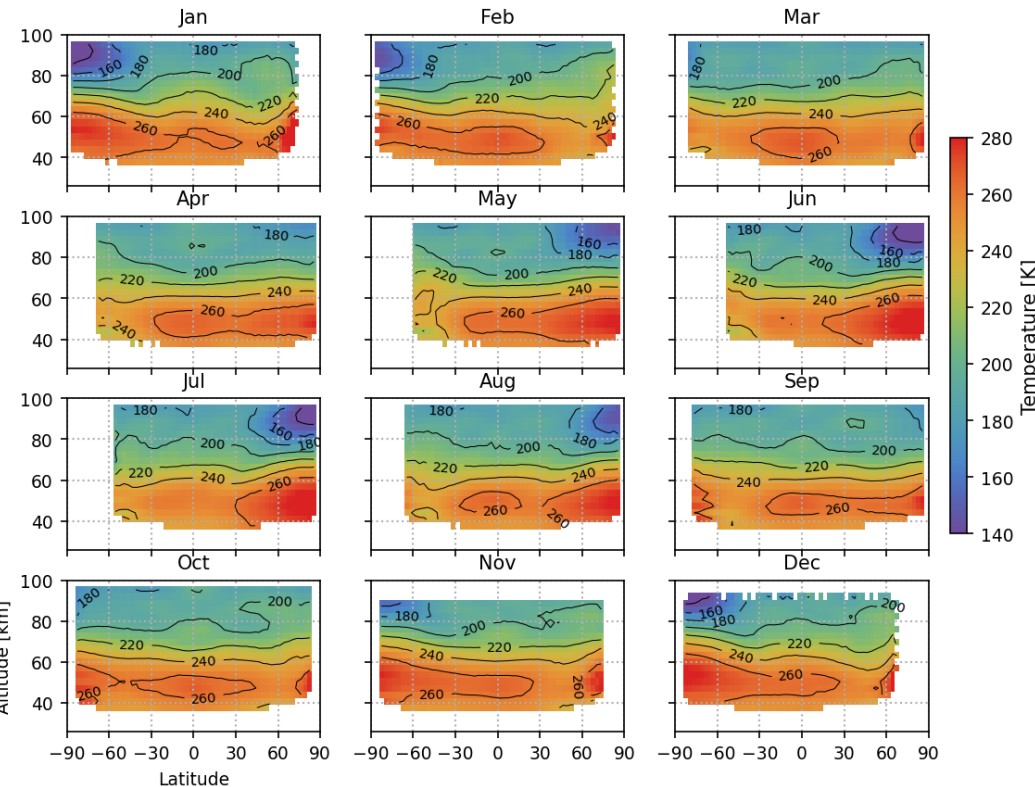

**Figure 12.** Retrieved temperature using $^1\Delta$ band airglow observed by nominal limb orbits. Only temperature points with DOFS greater than 0.5 are included in the averaging, and only grid boxes with more than 5 temperature points are shown.

temperature in the northern winter at high latitude and large SZA (generally larger than 90°). A closer examination of individual fits at those sounding locations reveals large residuals at lower tangent heights than 60 km, abnormally high $\chi^2$, and slow (if ever) convergence. The same region also has a large warm bias relative to ACE-FTS and MIPAS (to be shown in Section 5). We believe those are retrieval artifacts potentially due to horizontal gradient of airglow intensity at sunrise or sunset conditions.

## 435  5  Comparison of temperature profiles

Here we focus on comparing the retrieved temperature profiles with other instruments, as temperature is the most challenging part of the state vector to be retrieved. Reliable temperature retrievals will support the validity of the algorithm and product. An accurate upper atmospheric temperature climatology will also help better simulate the airglow spectral shapes.

## 5.1 Internal temperature comparison using MLT mode orbits

For the MLT mode, it is possible to intercompare temperature retrievals at the same observing locations using both the $O_2$ $^1\Delta$ and $A$ band airglow. Figure 13 compares temperature profiles retrieved by both bands in an MLT orbit (orbit # 41252 on 19 January 2010), colored by the altitude of retrieved layer. Panel (a) shows all data with DOFS larger than 0.1 over the full vertical range of profiles (roughly 50–120 km). The reddish points above 300 K are generally above 100 km and in the thermosphere. The other points below 300 K can be further classified into the mesopause region (80–100 km) and the mesosphere (below 80 km). Panel (b) zooms in at the mesopause region with a much higher DOFS threshold of 0.8 and shows a tight correlation (Pearson correlation coefficient $r = 0.91$) between $^1\Delta$ and $A$ band temperatures. As illustrated in retrieved profiles in Fig. 5 and Fig. 7, 80–100 km is the sweet spot where both $^1\Delta$ and $A$ band airglow are strong and generally free from self-absorption. Although the $^1\Delta$ and $A$ band retrievals use the same MSIS temperature as the prior, the information in the mesopause temperature is mostly from observations with high DOFS values. Out of the 504 temperature pairs with DOFS larger than 0.1 for both bands, 449 pairs have DOFS larger than 0.8. Below the mesopause region, the $A$ band-derived temperature becomes unreliable due to interference from self-absorption and loss of observational constraint. Over the full vertical range, the differences between those two temperatures are within the 95% confidence intervals labeled by the horizontal and vertical error bars in Fig. 13, which indicates that the posterior error generated by the optimal estimation-based algorithm adequately characterizes the uncertainty.

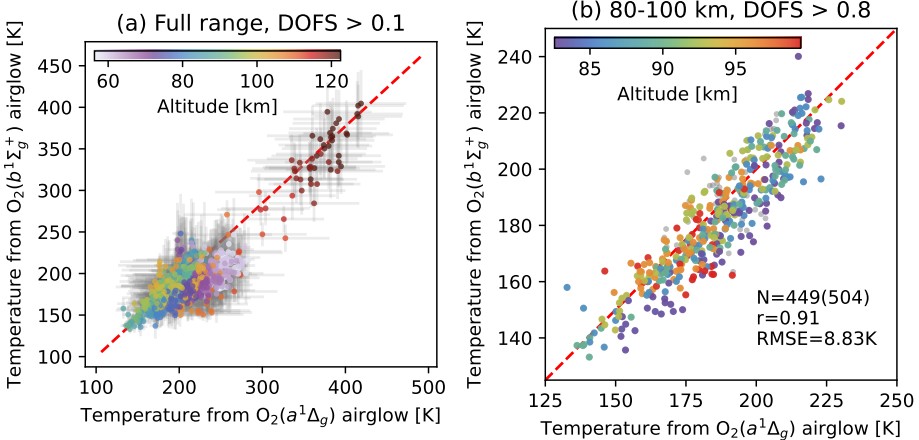

**Figure 13.** (a) Temperatures retrieved from both $^1\Delta$ and $A$ bands in an MLT mode orbit on 19 January 2010. Gray error bars indicate the 95% confidence intervals approximated by twice the posterior error standard deviation. Data are filtered by DOFS greater than 0.1 for both bands and colored by altitude. The red dashed line is 1:1. (b) is similar to (a) but only shows data between 80 km and 100 km. Data with DOFS greater than 0.1 but less than 0.8 are in gray (N=504). Data with DOFS greater than 0.8 are colored by altitude (N=449). Pearson correlation coefficient ($r$) and root mean squared error (RMSE) are also labeled.

Figure 14 extends the temperature comparison between $O_2$ $^1\Delta$ and $A$ bands over the mesopause region to all MLT orbits in each month of 2010. The MLT mode observations are conducted on two days in each month, so there are ~30 MLT orbits per month and 352 MLT orbits in 2010. The total number of data pairs, Pearson correlation coefficient, RMSE, and mean bias are labeled in the corresponding panel of each month. The mesopause temperature spans broader ranges in both polar winter and summer (January and December as well as June and July) than in other months due to the dynamics-driven cold summer polar mesopause. Overall, the correlation coefficients of mesopause temperatures from the two bands ranges from 0.75 to 0.90, with the $A$ band temperature colder than the $^1\Delta$ band temperature by 5–8 K. The low bias in the $A$ band temperature is likely caused by error propagated from lower altitude where retrieving temperature from $A$ band airglow becomes challenging due to strong self-absorption. The RMSE between the two bands is 10–12 K.

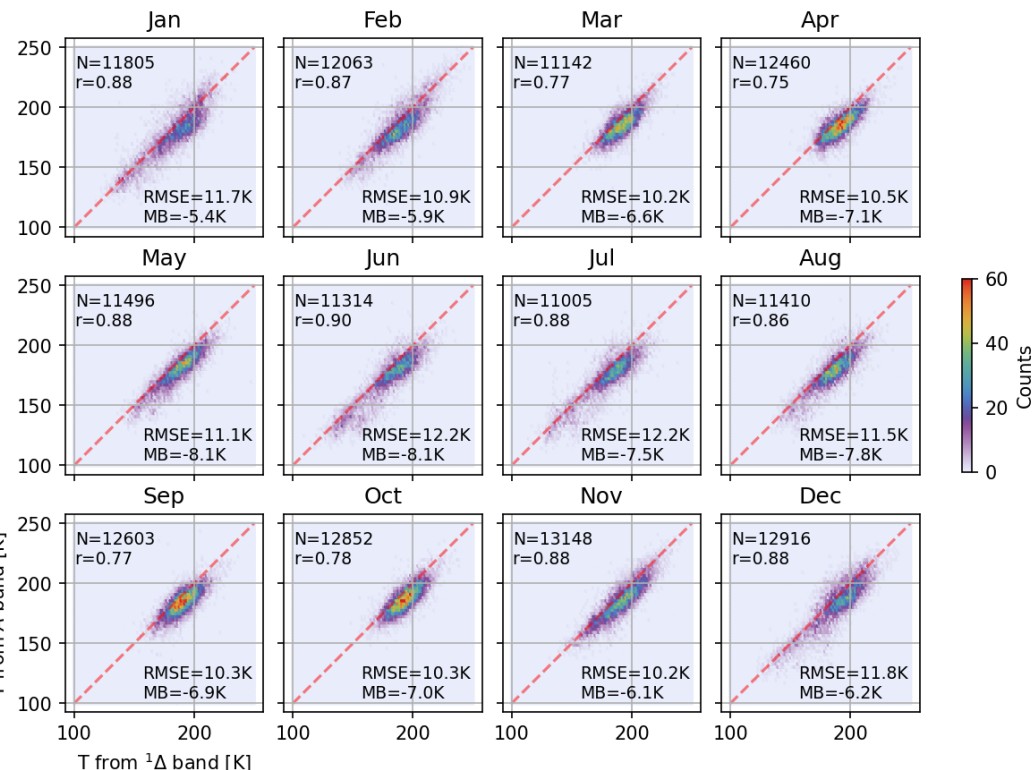

**Figure 14.** Comparisons between temperatures retrieved from $^1\Delta$ and $A$ band airglow over the mesopause region. Each panel include all MLT orbits in each month of 2010.

## 5.2 MLT and nominal limb temperature comparison with ACE-FTS

The ACE-FTS temperature retrieval covers the vertical ranges of airglow-derived temperature in both MLT and nominal modes of SCIAMACHY and can be compared with all temperature retrievals in this study. With coincidence criteria of 500 km and

2 hours, we identify ∼170 SCIAMACHY MLT vertical soundings and ∼1800 nominal vertical soundings that collocate with ACE-FTS soundings. The collocated soundings are sparse and skew toward mid to high latitudes (70°S–35°S and 42°N–70°N). Panels a and b of Fig. 15 display the mean bias and RMSE within 3.3-km vertical bins between the temperature profiles retrieved from $^1\Delta$ (pink) and $A$ (blue) bands relative to the collocated temperature profiles from ACE-FTS. The statistics for the prior temperature from MSIS are shown in green. The retrieved temperature data are limited to DOFS greater than 0.5. Compared with the prior MSIS temperature, both $^1\Delta$ and $A$ band-derived posterior temperatures demonstrate smaller RMSE relative to ACE-FTS, except at the high end (above 100 km for the $A$ band and above 95 km for the $^1\Delta$ band), where the airglow weakens, and at the low end (below 72 km) for the $A$ band, where the $O_2$ self-absorption strongly reduces observational constraints (cf. Fig. 7). The mean bias for the $^1\Delta$ band temperature stays generally less than $\pm 5$K below 90 km. In the mesopause region (80–100 km), the $A$ band temperature outperforms the $^1\Delta$ band temperature with lower absolute mean bias and RMSE relative to ACE-FTS. However, the quality of $A$ band temperature quickly degrades below 80 km. The relative mean biases are consistent with the internal airglow temperature comparisons in Section 5.1. To merge these two temperature retrievals from the MLT observations, one may use the $^1\Delta$ band-derived temperature below 80 km, use the $A$ band-derived temperature above 90 km, and average the two with linearly varying weights between 80 and 90 km.

A similar comparison is made between the $^1\Delta$ band derived temperature in the nominal limb orbits and ACE-FTS in Fig. 15c-d. The overall shapes of the mean biases of the $^1\Delta$ band temperature are consistent between panel a and c with the absolute mean bias in the nominal limb orbits slightly lower. Overall, the $^1\Delta$ band derived temperature agrees well with the ACE-FTS temperature (mean bias $< \pm 5$ K and significant reduction of RMSE relative to the prior) from 55 to 90 km. The substantial warm bias relative to ACE-FTS seen below 55 km originates from the retrieval artifact in northern high latitude winter and is overly amplified because the SCIAMACHY vs. ACE-FTS collocations coincidentally cluster in northern high latitude winter. A more clear picture will be depicted in the latitudinally resolved comparison with MIPAS in the following subsection. Results using HITRAN2016 spectroscopic data are shown as dotted lines in Fig. 15, while results using HITRAN2020 are in solid lines. Overall the $^1\Delta$ band retrievals become warmer and show more pronounced artifact near the northern high latitude stratopause when using HITRAN2020.

## 5.3 Nominal limb temperature comparison with MIPAS

Both MIPAS and SCIAMACHY were aboard the Envisat platform. Consequently, the MIPAS temperature soundings are spatiotemporally close to the SCIAMACHY soundings for all available orbits (see an example of one-day SCIAMACHY and MIPAS orbits in Fig. 8). Here we focus on the MIPAS comparison to $^1\Delta$ band-derived temperature for the nominal limb orbits. Figure 16 shows the latitudinally and vertically resolved mean bias between $O_2$ $^1\Delta$ band-derived temperature and MIPAS nominal mode temperature. The mean bias is calculated by subtracting the interpolated, same-orbit MIPAS temperature from each SCIAMACHY temperature profile and then regridding the resultant bias profile in the same way shown in Fig. 12. To avoid interpolation artifacts, we use the extended profiles in MIPAS Level 2 data, which fill the space above highest retrieval level by a seasonally and diurnally varying climatology. As a result, the comparison should be limited to below the top MIPAS nominal tangent height at 70 km. The mean bias is in general within $\pm 5$ K in the mesosphere and upper stratosphere throughout

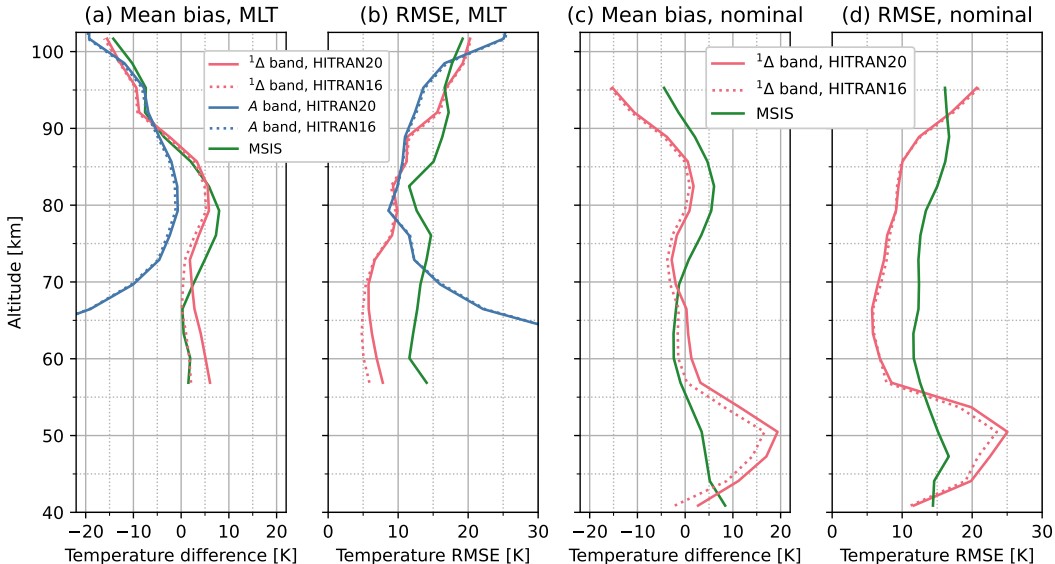

**Figure 15.** The first two panels show the mean bias (a) and RMSE (b) of $^1\Delta$ band (pink) and $A$ band (blue) temperature retrieved from the MLT orbits relative to the ACE-FTS collocated profiles. Panels c and d shows similar statistics between the $^1\Delta$ band temperature retrieved from nominal limb orbits and the ACE-FTS collocated profiles. Retrievals using HITRAN2016 spectroscopic data are shown as dotted lines vs. retrievals using HITRAN2020 in solid lines with the same color.

the year, and the airglow-derived temperature tends to be warmer than MIPAS. García-Comas et al. (2014) compared MIPAS with a range of satellite and ground-based temperature observations and found that MIPAS temperature differs from others by 2 K at 50–80 km in spring, autumn and winter at all latitudes, and summer at low to mid-latitudes. Differences between MIPAS and the other instruments in the summer high latitudes are typically smaller than 2 K at 50–65 km and 5 K at 65–80 km. MIPAS in general shows colder temperatures in the mid-mesosphere.

A consistent cold bias up to 10 K exists in the southern hemisphere stratopause region. As the DOFS for $^1\Delta$ band-derived temperature drops quickly below 50 km (Fig. 3), this bias stems mainly from a cold bias of the MSIS prior relative to MIPAS. The implausibly warm winter polar stratopause seen in Fig. 12 is more prominent in Fig. 16 with warm biases of over 20 K compared with MIPAS. Although it is clearly a retrieval artifact, we choose to keep those soundings because the temperature retrieval in the upper portion is still valid, and the retrieved emitting $O_2$ number densities are valuable to inform airglow distributions at large SZA. Figure 17 shows the RMSE between the $^1\Delta$ band temperature and MIPAS temperature. The RMSE values are largely below 10 K throughout the upper stratosphere and mesosphere with significant portions below 5 K. The exceptions are the aforementioned southern stratopause and northern polar winter stratopause biases. The fact that the RMSE values are similar to the absolute mean bias values indicates low variances in the retrieved $^1\Delta$ band temperature.

In addition, we compare the $O_2$ $^1\Delta$ band-derived temperature with the MIPAS IMK/IAA temperature product for the MA, UA, and NLC modes as shown in Fig. 18. The numbers of coincidence with SCIAMACHY for these modes are $\sim 20\%$ of

the MIPAS nominal mode, but they provide coverage above 70 km through the top of SCIAMACHY nominal mode retrieval. The SCIAMACHY-MIPAS temperature difference is consistent with the MIPAS nominal mode as in Fig. 16. The absolute temperature difference in the 70–100 km vertical range is generally within $\pm$ 5 K, except the summer mesopause at northern high latitude.

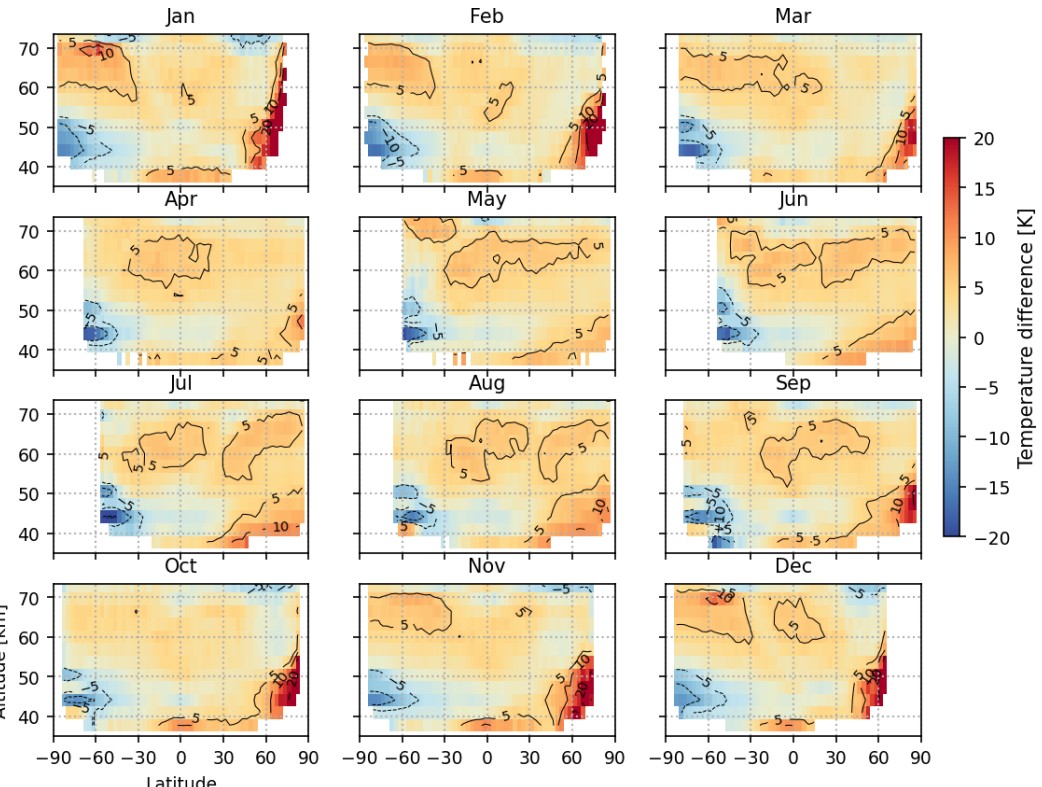

**Figure 16.** Monthly mean biases between temperature retrieved from $O_2$ $^1\Delta$ airglow in nominal limb orbits and the MIPAS nominal mode temperature. MIPAS temperature in the same orbit is first interpolated to the latitude and altitude of SCIAMACHY layers. The longitudinal difference is neglected. The comparison is limited to below 70 km, the maximum height for MIPAS nominal mode.

## 6   Conclusions and discussion

We develop an optimal estimation-based algorithm to retrieve $O_2$ $^1\Delta$ and $A$ band airglow as well as temperature ranging from upper stratosphere to lower thermosphere using limb-viewed spectra by the SCIAMACHY instrument. The algorithm is applied to nominal limb orbits that are available near daily to retrieve the $^1\Delta$ band airglow and MLT limb orbits that appear on two days per month to retrieve both $^1\Delta$ and $A$ band airglow in the year 2010. We demonstrate the monthly climatology for $O_2$ $^1\Delta$ band airglow and temperature retrieved from nominal limb observations, which will provide crucial information

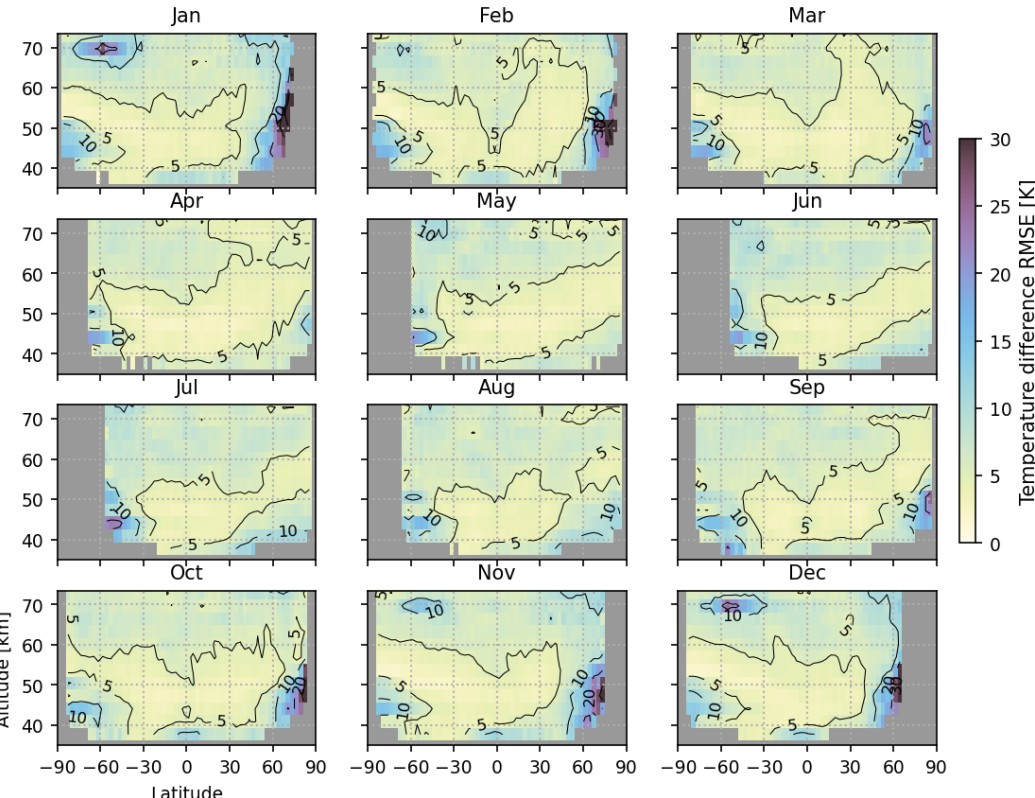

**Figure 17.** Monthly RMSE between temperature retrieved from $O_2$ $^1\Delta$ airglow in nominal limb orbits and the MIPAS temperature product.

for the consideration of airglow for future remote sensing of the $O_2$ $^1\Delta$ band. The global monthly distributions of the vertical column density of emitting $O_2$ in the $a^1\Delta_g$ state show mainly latitudinal dependence without other discernible geographical patterns. The $O_2$ $^1\Delta$ band-derived temperature agrees well with MIPAS Level 2 temperature with mean bias generally near or within $\pm$ 5K and RMSE below 10 K. Notable discrepancies include the stratospheric cold bias attributable to *a priori* influence, strong warm bias at winter polar stratopause region likely due to horizontal gradient of airglow intensity that violates the homogeneous layer assumption for the retrieval algorithm, and disagreements at the summer mesopause in northern high latitude. The reliable retrieval of temperature from airglow indicates that we can confidently reproduce the spectral shape of airglow emission.

The successful retrieval of the $O_2$ $A$ band airglow and the associated temperature demonstrates the generalizability of the airglow spectral model and the algorithm. The $A$ band and $^1\Delta$ band-derived temperature profiles are in good agreement in the mesopause region with the $A$ band-derived temperature lower by 5–8 K. Intercomparisons with ACE-FTS suggest that the $A$ band-derived temperature is of higher quality above $\sim$90 km due to the sharper decline of $^1\Delta$ band airglow with altitude. At lower altitude, the $^1\Delta$ band may provide more reliable temperature retrieval. Driven by the need of nadir observation of

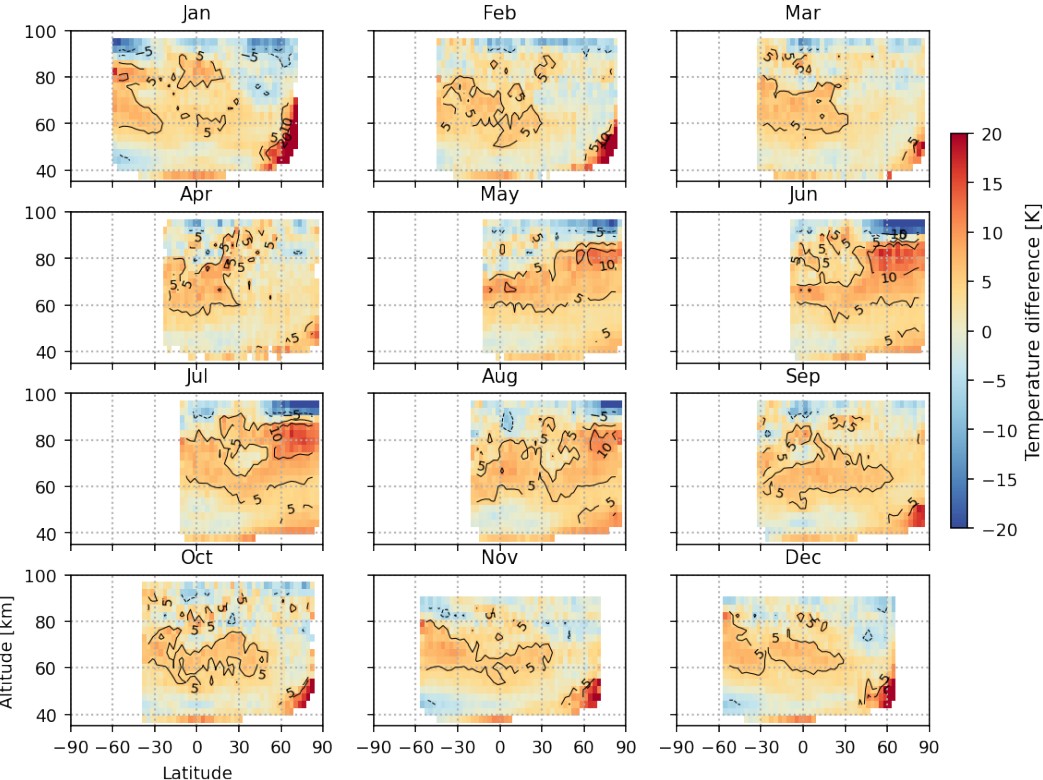

**Figure 18.** Similar to Fig. 16 but using MIPAS MA, UA, and NLC observation modes instead of nominal mode.

the $^1\Delta$ band by MethaneSAT, this study focuses on the $^1\Delta$ band retrieval, whereas the synergy between the $^1\Delta$ and $A$ band airglow may be further explored by future studies to improve understanding of the chemistry, radiation, and dynamics of the MLT region. It would also be interesting to include the Noxon band (Noxon, 1961) at 1.91 $\mu$m in future analyses. This band corresponds to the emission from the $b^1\Sigma_g$ to the $a^1\Delta_g$ excited electronic states and is another avenue for depopulation of the photo-chemically produced oxygen molecules in the $b^1\Sigma_g$ state. It is allowed through the electric quadrupole mechanism (Gordon et al., 2010).

The SCIAMACHY instrument made ten years of near daily nominal limb observations and four years of MLT mode limb observations. It is a natural future step to process the entire SCIAMACHY lifetime to generate long-term records of $O_2$ airglow emissions and upper atmospheric temperature. Comparison with MIPAS indicates that the current *a priori* temperature from the MSIS model may be inadequate, so it may be desirable to use more accurate reanalyzed temperature profiles where available and stitch the MSIS temperature above. A recent development improved the MSIS model especially at the MLT regions (Emmert et al., 2021), and future adoption of the new MSIS profiles will improve the *a priori* estimates. In contrast to previous studies using the SCIAMACHY limb observations, we do not aggregate across track, which leads to the options of keeping eight soundings across track or interweaving each pair of adjacent soundings into four soundings across track.

The retrievals in this study are conducted at the native across-track resolution and hence vertical resolution of 6.6 km. Future efforts will test the second option, which enhances the vertical resolution to 3.3 km, in line with previous SCIAMACHY limb studies (Bender et al., 2017; Zarboo et al., 2018).

*Code availability.* The underlying software code can be accessed at https://github.com/Kang-Sun-CfA/Methane/blob/master/l2_met/airglowOE.py.

## Appendix A: Derivation of emitting layer optical depth

Radiance at one end of the emitting segment with light path length $L$ and uniform absorber/emitter concentrations is

$$
\begin{aligned}
r_\lambda &= \frac{\varepsilon_\lambda}{4\pi} \int_0^L \exp\left(-[O_2]\sigma_\lambda(L-L')\right) \, dL' \\
&= \frac{\varepsilon_\lambda L}{4\pi} \left( \frac{1 - \exp(-[O_2]\sigma_\lambda L)}{[O_2]\sigma_\lambda L} \right).
\end{aligned} \tag{A1}
$$

The effective optical depth of the emitting segment ($\tilde{\tau}_\lambda$), assuming all emitters concentrated at the far end, should satisfy

$$
r_\lambda = \frac{\varepsilon_\lambda L}{4\pi} \exp(-\tilde{\tau}_\lambda) \tag{A2}
$$

The optical depth of this segment as a transmitting layer is simply

$$
\tau_\lambda = [O_2]\sigma_\lambda L. \tag{A3}
$$

Combining Eq. A1, A2, and A3:

$$
\tilde{\tau}_\lambda = -\ln\left( \frac{1 - \exp(-\tau_\lambda)}{\tau_\lambda} \right). \tag{A4}
$$

This is equivalent to Eq. 17. At optical thin limit, applying L'Hôpital's rule

$$
\begin{aligned}
\lim_{\tau_\lambda \to 0} \frac{\tilde{\tau}_\lambda}{\tau_\lambda} &= \lim_{\tau_\lambda \to 0} \frac{-\ln\left(\frac{1 - \exp(-\tau_\lambda)}{\tau_\lambda}\right)}{\tau_\lambda} \\
&= \lim_{\tau_\lambda \to 0} \frac{\tau_\lambda \exp(-\tau_\lambda) + \exp(-\tau_\lambda) - 1}{\tau_\lambda(\exp(-\tau_\lambda) - 1)} \\
&= \lim_{\tau_\lambda \to 0} \frac{-\exp(-\tau_\lambda) + \tau_\lambda \exp(-\tau_\lambda)}{-2\exp(-\tau_\lambda) + \tau_\lambda \exp(-\tau_\lambda)} \\
&= \frac{1}{2}.
\end{aligned} \tag{A5}
$$

Namely, the effective optical depth of the emitting segment due to the self absorption is half of its optical depth as a transmitting segment at optical thin limit, which makes intuitive sense. At the optical thick limit, the effective optical depth of emitting segment due to self absorption approaches the natural logarithm of its optical depth as a transmitting segment:

$$
\lim_{\tau_\lambda \to \infty} \tilde{\tau}_\lambda = \ln \tau_\lambda. \tag{A6}
$$

## Appendix B: Jacobians of limb-viewed radiance with respect to temperature, $[O_2^*]$, and $\ln([O_2])$

The Jacobian of radiance observed at limb view $i$ with respect to the temperature of layer $j$ ($j \geq i$, otherwise the Jacobian values are all zero) is

$$
\frac{\partial r_{\lambda,i}}{\partial T_j} = \sum_{l \in \{N+j-i, N-j+1\}} \left( \frac{L_{ij}}{4\pi} \frac{\partial \varepsilon_{\lambda,j}}{\partial T_j} \exp\left( -\tilde{\tau}_{\lambda,ij} - \sum_{l'=1}^{l-1} \tau_{\lambda,ij'} \right) \right)
$$

$$
+ \sum_{l \in \{N+j-i, N-j+1\}} \left( \frac{L_{ij}\varepsilon_{\lambda,j}}{4\pi} \left( -\frac{\partial \tilde{\tau}_{\lambda,ij}}{\partial T_j} \right) \exp\left( -\tilde{\tau}_{\lambda,ij} - \sum_{l'=1}^{l-1} \tau_{\lambda,ij'} \right) \right)
$$

$$
+ \sum_{l \in \{N+j-i, N-j+1\}} \sum_{l'=l+1}^{2N-i} \left( \frac{L_{ij'}\varepsilon_{\lambda,j'}}{4\pi} \left( -\frac{\partial \tau_{\lambda,ij}}{\partial T_j} \right) \exp\left( -\tilde{\tau}_{\lambda,ij'} - \sum_{l''=1}^{l'-1} \tau_{\lambda,ij''} \right) \right). \tag{B1}
$$

Here the first term on the right-hand side reflects the sensitivity of airglow emission to the temperature of layer $j$ (with two corresponding segments as in Eq. 14); the second term reflects the temperature sensitivity of self-absorption of layer $j$ as an emitting layer; and the third term reflects the temperature sensitivity of layer $j$ as a transmitting layer, where temperature alters its absorption from upstream emissions.

The temperature sensitivities of both a layer's transmitting optical depth ($\partial \tau_\lambda / \partial T$) and its effective optical depth as an emitting layer ($\partial \tilde{\tau}_\lambda / \partial T$) can be related to the temperature sensitivity of absorption cross section ($\partial \sigma_\lambda / \partial T$, obtained by finite difference using HAPI) by applying the chain rule to Eq. 16 and 17. We assume that the ground-state $O_2$ number density $[O_2]$ does not depend on temperature and retrieve $[O_2]$ profile independently to account for errors in *a priori* $O_2$ profile. The temperature sensitivity of emissivity ($\partial \varepsilon_\lambda / \partial T$) is given by Eq. 10. The derived temperature Jacobian in Eq. B1 is confirmed with numerical finite difference derivative of Eq. 15 at the instrument resolution and sampling grid. The comparison is shown in Fig. B1. The analytical Jacobians are consistent with finite difference within $10^{-6}$ for all tangent heights.

The Jacobians of emitting $O_2$ number density are derived similarly but only involving the emitting segment:

$$
\frac{\partial r_{\lambda,i}}{\partial [O_2^*]_j} = \sum_{l \in \{N+j-i, N-j+1\}} \left( \frac{L_{ij}}{4\pi} \frac{\partial \varepsilon_{\lambda,j}}{\partial [O_2^*]_j} \exp\left( -\tilde{\tau}_{\lambda,ij} - \sum_{l'=1}^{l-1} \tau_{\lambda,ij'} \right) \right), \tag{B2}
$$

where $\partial \varepsilon_{\lambda,j} / \partial [O_2^*]_j$ is the Jacobian of airglow emissivity with respect to $[O_2^*]$ for layer $j$ and given by equation 9.

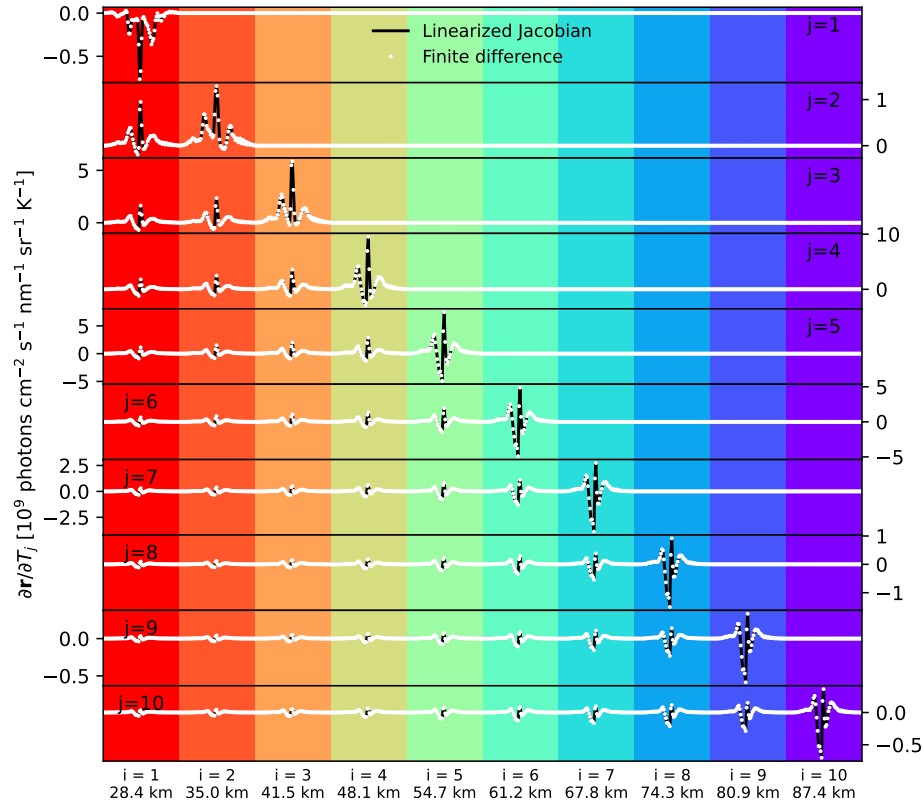

**Figure B1.** Validation of analytical Jacobians of radiance with respect to temperature based on Eq. B1 (black lines) with finite difference (white dots). Each vertical column represents one limb view (fixed $i$) over all layers (different $j$). Ten tangent views (i.e., $N = 10$) are included using a single vertical sounding at the $O_2$ $^1\Delta$ band from SCIAMACHY orbit # 41011 on 3 January 2010. Only Jacobians of layers above the tangent point are non-zero.

Besides, we also include relative changes to the ground-state $O_2$ number density in the state vector, which is equivalent to retrieving the natural logarithm of ground-state $O_2$ number density, $\ln([O_2])$. The corresponding Jacobians are given by

$$
\begin{aligned}
\frac{\partial r_{\lambda,i}}{\partial \ln([O_2]_j)} =& [O_2]_j \frac{\partial r_{\lambda,i}}{\partial [O_2]_j} \\
=& [O_2]_j \sum_{l \in \{N+j-i, N-j+1\}} \left( \frac{L_{ij}\varepsilon_{\lambda,j}}{4\pi} \left( -\frac{\partial \tilde{\tau}_{\lambda,ij}}{\partial [O_2]_j} \right) \exp\left( -\tilde{\tau}_{\lambda,ij} - \sum_{l'=1}^{l-1} \tau_{\lambda,ij'} \right) \right) \\
& + [O_2]_j \sum_{l \in \{N+j-i, N-j+1\}} \sum_{l'=l+1}^{2N-i} \left( \frac{L_{ij'}\varepsilon_{\lambda,j'}}{4\pi} \left( -\frac{\partial \tau_{\lambda,ij}}{\partial [O_2]_j} \right) \exp\left( -\tilde{\tau}_{\lambda,ij'} - \sum_{l''=1}^{l'-1} \tau_{\lambda,ij''} \right) \right).
\end{aligned} \tag{B3}
$$

Here the first term on the right hand side of Eq. B3 reflects the sensitivity of the effective optical depth of layer $j$ to its own $O_2$ number density, and the second term reflects the sensitivity of its optical depth to its $O_2$ number density as a transmitting layer.

Both $\partial \tau_\lambda / \partial [O_2])$ and $\partial \tilde{\tau}_\lambda / \partial [O_2]$ can be readily calculated by differentiating Eq. 16 and 17.

*Author contributions.* GGA and EO'S generated the SCIAMACHY Level 1 files. KS developed and implemented the forward model and retrieval algorithm with inputs from XL and CCM. IEG provided expertise in spectroscopy. CES provided expertise in upper atmospheric chemistry and observations and helped with scientific discussion and interpretation. XL, SCW, KC, and KS managed the project. KS and MY performed the temperature intercomparison and wrote the manuscript. All coauthors contributed to the editing of the manuscript.

*Competing interests.* The authors declare that they have no conflict of interest.

*Acknowledgements.* We acknowledge MethaneSAT, LLC and the Environmental Defense Fund for supporting science and algorithms at the University at Buffalo and the Smithsonian Astrophysical Observatory. We thank Roman V. Kochanov for the help with HAPI. The SCIA-MACHY Level 1 data processing is supported by NASA Making Earth System Data Records for Use in Research Environments program (grant number 80NSSC18M0091). Some of the computations in this paper were conducted on the Center for Computational Research at the

University at Buffalo (UB CCR, 2022) and the Smithsonian High Performance Cluster, Smithsonian Institution (SI HPC, 2022).

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
