# Peer review of "An optimal estimation-based retrieval of upper atmospheric oxygen airglow and temperature from SCIAMACHY limb observations"

_Atmospheric Measurement Techniques, 2022_

## Author Comment (AC1)

Response to Referee #1:

We appreciate the very helpful feedback from the referee. The referee's comments are listed in *italics*, followed by our response in blue. New/modified text in the manuscript is in **bold**.

*In this paper, a new retrieval algorithm for temperature and O2 VER is introduced for the O2(1Delta) and O2(1Sigma) bands measured by Sciamachy. O2 VER and temperatures have been derived from these observations before; what is new here is that both are derived simultaneously, and self-absorption is considered in a consistent. The retrieval is applied to one year of data (2010), and temperature data are compared to ACE-FTS and Mipas. The Mipas comparison is particularly useful as Mipas was on the same satellite as Sciamachy, therefore providing close coincidences. The O2 airglow is highly relevant both for the accuracy of greenhouse gas remote sensing products, and for the energy budget of the mesosphere / lower thermosphere, and the data from this new algorithm provide a large step forward compared to previous publications. The paper is also generally very well written. However, I have some questions e.g., regarding the derivation of the prior error and the comparison to Mipas MA/UA data, as well as a few minor points listed below.*

Thanks for the comments. Note we change number density notation from n[*] to [*] following the suggestion from referee#2. The description about prior error is clarified, and MIPAS MA/UA/NLC modes are compared with SCIAMACHY [1]$\Delta$ band airglow temperature. See the responses below.

*Line 249-250: Doesn't this imply an altitude dependent differently strong weighting, as the self-absorption affects the lower levels exponentially stronger?*

Self-absorption indeed causes inaccurate results in the linear inversion. It is accounted for in the nonlinear retrieval system in this study. See the following response about the weighting between information from the prior and from observations.

*Line 251-252: The statement that a prior error of 100 times the prior value leads to a weak to negligible prior constraint seems not correct in the lower altitudes affected by self-absorption: as there the prior profile is too low, and might be orders-of-magnitude too low, so is the prior error actually quite low. A climatology might be a better estimate of the prior values here, if available.*

To the best of our knowledge, there is no available climatology for emission [1]$\Delta$ $O_2$, except results above 60 km based on linear inversion (self-absorption not considered) from Zarboo et al. (2018). Actually, the prior profile is too high, not too low, in the stratosphere where self-absorption is significant. This is because we use a constant value for all altitudes in the prior profile, as indicated in lines 248-249:

"For each retrieval, we first conduct a linear inversion of the spectra and use the vertical mean value of the inverted $[O_2^*]$ profile as the prior values for the $[O_2^*]$ profile."

Therefore, the prior error is also a constant (100 times larger than the prior value, which is constant at all altitudes) and much larger than any possible $[O_2^*]$ values. Figure 3c, figure 5c,

and figure 7c also show that the DOFS of [O$_2$*] are effectively one, supporting negligible prior constraint. To clarify, lines 251-252 are revised to

**"The corresponding prior error is set to be 100 times the prior, a constant value for all altitudes. This effectively gives no prior constraint to the [O$_2$*] profile and assures its information all comes from observations through near-unity DOFS of retrieved [O$_2$*]."**

*Line 355: These missing points … are these related to high solar zenith angles? As during daytime the dominating formation mechanism is O3 absorption, the O2 airglow varies strongly from daytime to nighttime, and observations with high SZAs would provide very different (lower) values, the signal-to-noise is also low. This should be discussed somewhere, as you don't separate daytime and nighttime observations at high latitudes, and it should also be stressed in discussing your climatology of O2 airglow: it is a climatology covering a whole year of observations, but at a very specific time of day, about 10:00 local solar time.*

Yes, these missing points are mostly at high solar zenith angles and the transition zone between day and night. Note we remove the ascending portion and SZA above 100° when averaging individual orbits into the climatology. This removes some ambiguous twilight data in polar regions. The sentences at lines 355-359 are revised to

**"These missing points are generally located at high latitudes and high solar zenith angles. In these transition regions between daytime and nighttime, the horizontal variation of airglow intensity is significant, which violates the homogeneous layer assumption for the retrieval algorithm. Retrieved data are often available for part of the ascending phase of the orbit at the summer hemisphere (most valid data are in the descending phase), leading to some repeated observations at the same latitude, although at different SZAs and potentially in the nighttime. To eliminate such a latitudinal ambiguity and nighttime data, we remove the ascending portion when averaging over multiple days and limit the SZA to within 100°."**

The last sentence in the introduction is revised to

**"The algorithm is applied to one year of SCIAMACHY of limb observations, including the MLT mode, to construct a climatology of O$_2$ airglow and upper atmospheric temperature at 10:00 local solar time."**

The sentence at lines 81-82 is updated to

**"The instrument was launched on board the Envisat satellite which was operational on a sun synchronous orbit with an equator crossing time in the descending node of 10:00 local solar time from March 2002 until April 2012."**

*Line 437: can you provide some idea why the A band has such a stable cold bias compared to the 1Delta?*

The systematic difference between temperatures derived from the two bands is more clearly shown in Figure 15a. One possible reason is the propagation of temperature errors from lower altitudes in the A band, where strong self-absorption leads to diminishing observable emission

signals. Spectroscopic errors may also play a role, given the slight difference between HITRAN16 and HITRAN20. One sentence is added

**"The low bias in the *A* band temperature is likely caused by error propagated from lower altitude where retrieving temperature from *A* band airglow becomes challenging due to strong self-absorption."**

*Line 468: "Mipas temperature retrieval in 2010 is only available below ~80 km": This statement is factually not correct. A) there are the middle atmosphere / upper atmosphere limb modes of MIPAS which scan up to 120 km respectively 170 km every ten days since 2007. These were coordinated with the Sciamachy MLT mode in such a way that corresponding observations are available every 30 days – about once per months. Observations in the MA/UA modes were carried out also in 2010, and temperatures were retrieved from these modes up to at least 120 km, see e.g., Fig 4 in Sinnhuber et al, JGR, 2022 for an example. Data are available on the MIPAS data server at IMK (https://www.imk-asf.kit.edu/english/308.php), and I am sure the Mipas team (e.g. Bernd Funke or Thomas von Clarmann for the MA/UA modes) would be happy to help in accessing and applying the data. If there are coincidence data between Sciamachy and Mipas for 2010 (and there should be at least 12 days) please do the comparison. B) Just as a caution, the nominal limb mode of Mipas scans up to 68 km, so values above ~70 km are probably dominated by the prior profile.*

We add the discussion of the additional MIPAS mode in Section 2.4:

**"The measurement modes of MIPAS used in this study include the nominal measurement mode with an altitude coverage of roughly 6-70 km, the middle atmosphere (MA) mode covering 18-102 km, the upper atmospheric (UA) mode covering 42-172 km, and the noctilucent cloud (NLC) mode covering 39-102 km. The nominal measurement mode makes up the bulk of MIPAS measurements, whereas the MA and UA modes were available every at least 10 days, and the NLC mode only happened on a few days in 2010. We use the nominal temperature profiles from version 8 of MIPAS Level 2 data retrieved by ESA (Dinelli et al., 2021). Version 8 data from the other modes are obtained through the Institute of Meteorology and Climate Research in cooperation with the Instituto de Astrofísica de Andalucía (IMK/IAA) retrieval algorithm (García-Comas et al., 2012; Kiefer et al., 2021). The typical total errors are 0.5-2 K below 70 km and 2-7 K above (for MA, UA, and NLC modes). The typical vertical resolutions in the comparison range of this study are 3-7 km."**

The sentence at line 468 is removed. Figures 16-17 are changed to only show up to ~70 km, and the text is revised to reflect that

**"To avoid interpolation artifacts, we use the extended profiles in MIPAS Level 2 data, which fill the space above highest retrieval level by a seasonally and diurnally varying climatology. As a result, the comparison should be limited to below the top MIPAS nominal tangent height at 70 km."**

Comparison between SCIAMACHY $^1\Delta$ band-derived temperature in nominal mode with MIPAS MA, UA, and NLC modes are included at the end of this section. We will leave a more complete intercomparison/validation involving the SCIAMCHY MLT mode to future studies, as this manuscript will not focus on temperature validation.

**"In addition, we compare the O₂ $^1\Delta$ band-derived temperature with the MIPAS IMK/IAA temperature product for the MA, UA, and NLC modes as shown in Fig. 18. The numbers of coincidence with SCIAMACHY for these modes are ~20% of the MIPAS nominal mode, but they provide coverage above 70 km through the top of SCIAMACHY nominal mode retrieval. The SCIAMACHY-MIPAS temperature difference is consistent with the MIPAS nominal mode as in Fig. 16. The absolute temperature difference in the 70--100 km vertical range is generally within ±5 K, except the summer mesopause at northern high latitude.**

[Figure]

**Figure 18. Similar to Fig. 16 but using MIPAS MA, UA, and NLC observation modes instead of nominal mode.**

"

*Figure 16: Here Mipas temperatures are used up to nearly 100 km – if they are from the nominal mode as you imply, the large differences above 80 km are to be expected, as the nominal mode scans up to 68 km only. It's rather surprising that the region 70-80 km seems to agree fairly well in most month.*

We limit the comparison with MIPAS nominal data to below 70 km and add MA/UA/NLC modes to show above 70 km. See the previous response.

*Minor points:*

*Abstract: I know they are commonly used, but nevertheless I found the use of the abbreviations (1Δ and A) for the bands slightly irritating. Could you use the full names (O2(a1Δg), O2(b1Σg+) at least in the abstract?*

Thank you for the suggestion. We changed to the full names in the abstract.

*Line 5: as the nominal mode only scans up to 93 km in 2010, how do you derive O2(1Delta) in 93-100 km?*

93 km is the tangent height, but we retrieve layer properties using tangent height as the lower boundary of each layer. The top layer is assumed to be between the top tangent height and top tangent height plus average layer thickness, which is about 6.6 km. This adds up to about 100 km. The layering scheme is introduced in lines 137-139 in the manuscript:

"**An atmospheric layer bounded by two tangent heights of SCIAMACHY limb observations is the basic spatial resolving unit of this study. We create an additional layer above the outermost tangent height by assuming a layer thickness equal to the average difference between adjacent tangent heights.**"

*Lines 9 – 11: please add altitude range where temperatures can be retrieved (~40 – 95 km for nominal mode, 65 – 105 km for the MLT mode?).*

We added the valid altitude range of 40-100 km for the nominal and 60-105 km for the MLT mode, combining both bands.

*Line 62: Yang et al, SPECTROSCOPY AND SPECTRAL ANALYSIS, 2021 also used the O2 airglow to derive temperatures*

It is added to the references.

*Line 84: in the nominal limb mode, Sciamachy scanned up to 93 km in 2010. It was slightly higher at the beginning of the Sciamachys operations, but unfortunately this was changed to 93 km already in late 2003.*

It is updated to "**In nominal limb mode, SCIAMACHY observed the atmosphere from the surface up to 93 km in 2010.**"

*Line 166: where is the number 1.4387760 cm K coming from?*

It is the second radiation constant in Planck's law, and its value equals to the produce of Planck's constant, speed of light, and the inverse of the Boltzmann constant (e.g., see https://hitran.org/docs/definitions-and-units/). This sentence is updated to "**where $c_0$ is a scaling constant, and $c_2$ is the second radiation constant in Planck's law with a value of 1.4387769 cm K.**"

*Line 192: ... will also be N. Actually if you formulate it like that, the number should be N-1. It is N in your retrieval because you add an upper bound layer at the top. Can you clarify this?*

In fact we create a layer for each tangent height, so the number of layers equals the number of tangent heights and equals N. We clarify this by modifying lines 137-139 to

**"An atmospheric layer bounded by two tangent heights of SCIAMACHY limb observations is the basic spatial resolving unit of this study. We create an additional layer above the outermost tangent height by assuming a layer thickness equal to the average difference between adjacent tangent heights."**

*Line 300-302: "only limited limb views with deeper tangent heights could observe those layers" I am not quite sure I understand this statement. Does it mean only some of the nominal limb scans (which all go down to the surface) provide a good signal-to-noise ratio in these altitudes? This is how I understood this sentence, however I don't understand how it applies to the discussion of a single limb profile as given here. Please clarify.*

"Deeper" here is relative in the collection of tangent heights above ~25 km and does not involve limb scans down to the surface. All tangent views can see the top layer, whereas only the bottom tangent view can see the bottom layer. To clarify, item (1) is rewritten:

**"The retrieved [$O_2(a^1\Delta_g)$] becomes increasingly uncertain down to the stratosphere because (1) the lower the layer is, the smaller the number of tangent views can detect them…"**

*Line 303: as supported by comparison to results of the MLT mode retrieval*

Revised as suggested.

*Lines 311-312: This is by design … due to the self-absorption*

Revised as suggested.

*Lines 317-318: a) the lowest tangent altitude of the MLT mode is around 51 km; b) why is the upper limit set to below 120 km?*

a) This particular sounding only goes to 57 km. This relates to our choice of using 8 native across-track soundings, instead of combining adjacent soundings into 4 across-track soundings.

b) This is because above 120 km effectively no $O_2(a^1\Delta_g)$ airglow is observed, and a cut-off saves computing time. Similarly, the A band is cut off at 130 km. This sentence is revised to

**"The lowest tangent height of this particular MLT sounding is at 57 km, and the upper limit is set to below 120 km, above which the airglow is negligible."**

*Line 324: erase the would. They do.*

Revised as suggested.

*Line 371: the maximum abundance*

Revised as suggested.

---

## Author Comment (AC2)

Response to Referee #2:

We appreciate the very helpful feedback from the referee. The referee's comments are listed in *italics*, followed by our response in blue. New/modified text in the manuscript is in **bold**.

*This was a straightforward, well written paper that I enjoyed reading. For the most part, the methodology and results were clearly presented, although clarification is needed in a few spots (as listed below). After properly addressing the following (mostly minor) issues, I'd recommend the paper for publication.*

*"1Δ" should be "$^1Δ$" throughout the manuscript*

Revised as suggested.

*line 8 – I'm not sure I've ever heard the term "loading" before in the context of airglow. If this is a common term and I just haven't been paying attention, then it's fine, but otherwise I would suggest the word "density" or "concentration" instead.*

Revised to "density" as suggested.

*line 35 – Please briefly explain in the text what is meant by "confounds the retrieval algorithm"*

This sentence is expanded to

"**The $O_2$ $A$ band also overlaps with strong terrestrial solar-induced fluorescence that provides valuable information on plant photosynthesis but perturbs the $O_2$ absorption features and, if not properly accounted for, leads to systematically biased greenhouse gas retrievals (Frankenberg et al. 2012).**"

*36 – "As an alternative" confused me, as you've already talked about the 1Δ band. I'd suggest you don't need an introductory remark for this sentence*

Removed as suggested.

*40-41 – Please be specific about what there is a "lack of assessment" of*

Revised to "… **but there is a lack of quantitative assessment of how the $A$ band airglow impacts greenhouse gas retrieval.**"

*42 – I think "chemically" should be "photochemically"*

Revised as suggested.

*53 – Please specify what is meant by "subject to errors"*

"Errors" is replaced by "**systematic biases in the retrieved airglow VER**", and the reference to Sun et al. (2018) GRL is added. The bias is illustrated by Figure 2b in that paper (note the negative VER bias from linear inversion below 60 km):

[Figure]

*70 – Based on your discussion of the results I don't think "degrees of freedom of signal" is the correct term. As per Rodgers, DOFS is trace(A), which is a single value. It looks to me like what you're calling DOFS is what is more commonly referred to as the retrieval response, i.e. the sum of the rows of A. Please clear this up.*

Thanks for the comment. We use the DOFS as the sum of diagonal elements of the averaging kernel (AK) matrix, but not necessarily the entire AK. For example, an algorithm could retrieve a 50-element state vector where 10 elements are from a profile. It is not uncommon to sum the diagonal elements of that 10 state vector elements and report the DOFS of the profile retrieval. Sometimes DOFS in sub profiles is reported. For example, Kulawik et al. (2017, https://acp.copernicus.org/articles/17/5407/2017/acp-17-5407-2017.pdf) separated the GOSAT $CO_2$ retrieval of 1.6 DOFS into two partial columns with 0.8 DOFS each. The individual diagonal elements of AK may be referred to as "averaging kernel sensitivity" (e.g., Lu et al. 2022, https://acp.copernicus.org/articles/22/395/2022/acp-22-395-2022.html), but we prefer not introducing a new term when the concept of DOFS can be made inclusive. This is also consistent with the ozone profile work by Liu et al. (2010, https://acp.copernicus.org/articles/10/2521/2010/acp-10-2521-2010.pdf).

We modify this sentence to

"**The use of Bayesian inversion enables incorporation of *a priori* knowledge, balancing of measurement error and prior error, and detailed posterior error analysis including the averaging kernel matrix and degrees of freedom for signal (DOFS) (Rodgers, 2000, Brasseur and Jacob, 2017).**"

The definition of DOFS is given at line 274

"**In this study, the DOFS of each state vector element refers to the corresponding diagonal element in the averaging kernel (Liu et al., 2010).**"

*71 – please specify what is meant by "the formula"*

It refers to equations 3-6 in this manuscript. We modify this sentence and leave detailed explanation to section 3:

**"The airglow emission spectra are simulated based on the spectral model for the $O_2$ $^1\Delta$ band proposed by Sun et al. (2018), which we demonstrate can be extended to the $O_2$ $A$ band with simple generalization."**

*84 – "tangentially" is not needed*

Removed as suggested.

*87 – file format details are unnecessary*

Due to the complexity of SCIAMACHY native file format, this is actually a significant amount of work that is crucial for the following retrieval. The file formats and the usage of the SciaL1C command-line tool are frequently mentioned in SCIMACHY retrieval papers. Our customized algorithm converted the L1B files to a much more user-friendly NetCDF format, and more importantly, enabled separation of 8 across track positions instead of always averaging them. Therefore, we would prefer keeping these sentences.

*99-104 – I would assume that a proper background correction would be critical in altitude regions far from the emission profile peak, so I think this requires a bit more discussion. It would also be interesting to see a plot of typical or averaged background signals. Whether or not you include a plot of the background signals, I would appreciate at least a brief discussion on the shape of the background signals, and the assumptions/limitations that go in to using a scaled thermospheric signal.*

The background correction follows Zarboo et al. (2018). Figure 2 of that paper, copied below, shows uncorrected and corrected limb radiance for both bands. The spectrum at 142 km is a typical background signal for the $A$ band, whereas the background for the $^1\Delta$ band is simply a linear fit using out-of-band radiances.

[Figure]

**Figure 2.** Examples of the daytime-calibrated spectra and the background-corrected spectra. **(a)** $O_2(^1\Sigma)$ on 3 February 2010; orbit 41 455; mean latitude 17.3° N, mean longitude 94.3° E. **(b)** as **(a)** but for the $O_2(^1\Delta)$ band. **(c)** as **(a)** but with background correction applied. **(d)** as **(b)** but with background correction applied.

We expand the discussion:

**"In addition to airglow emissions, the observed radiance spectra contain photons from Rayleigh scattering and multiple scattering by the atmosphere and the surface. The scattering signal can be approximated by limb views at maximum tangent heights and contains $O_2$ absorption feature for the $A$ band. No $O_2$ absorption is observed for the $^1\Delta$ band due to much weaker atmospheric scattering and lower $O_2$ absorption. To account for the scattered light, a background signal consisting of averaged $A$ band spectra at high tangent heights (130-150 km) from the same sounding is subtracted from each limb spectrum (Zarboo et al., 2018). Before each subtraction, the background signal is scaled to match its out-of-band radiance with the out-of-band radiance of the limb spectrum to be corrected for. This step assures the out-of-band radiance centers at zero after correction. This correction assumes the spectral shape of scattered light is the same for all tangent heights and may leads to systematic errors at low tangent heights from the $A$ band, where airglow emission is low, and scattering path may differ significantly from the thermosphere. Following Zarboo et al. (2018), we consider 750-759 nm and 767-780 nm to be out of band."**

*Section 2.2 – MSIS v2 is now the most recent version, https://doi.org/10.1029/2020EA001321. I don't necessarily think you need to update to this version (although it would definitely strengthen the paper), but, if you don't, you at least need to discuss the limitations of MSIS-E-00 (especially in polar MLT regions) as is done in some of your references.*

Thanks for the information. We rely on a third-party Python package to interface MSIS, and there seems to be no easy way to update. We referred to the MSIS v2 paper in the last paragraph as a future direction:

**"A recent development improved the MSIS model especially at the MLT regions (Emmert et al., 2020), and future adoption of the new MSIS profiles will improve the *a priori* estimates."**

*118 – "into two high and low altitude regimes" should just be "into two altitude regimes"*

Revised as suggested.

*Section 2.3 – temperatures from ACE-FTS have been used in multiple comparison studies (easily found at https://ace.scisat.ca/publications/). Please briefly discuss the results of these studies so readers have an idea of the quality of the ACE-FTS temperatures in the upper atmosphere.*

The following sentences are added to Section 2.3:

**"The ACE-FTS temperature profiles have been extensively used in the validation of profiles from other instruments such as SOFIE (Marshall et al., 2011), MIPAS (García-Comas et al., 2014), and OSIRIS (Sheese et al., 2012)."**

*Section 2.4 – same as ACE-FTS, please briefly discuss the quality of MIPAS temperature retrievals*

We expand the discussion about MIPAS in Section 2.4. Specifically, the MA/UA/NLC modes are added according to a suggestion from referee#1:

"**The measurement modes of MIPAS used in this study include the nominal measurement mode with an altitude coverage of roughly 6-70 km, the middle atmosphere (MA) mode covering 18-102 km, the upper atmospheric (UA) mode covering 42-172 km, and the noctilucent cloud (NLC) mode covering 39-102 km. The nominal measurement mode makes up the bulk of MIPAS measurements, whereas the MA and UA modes were available every at least 10 days, and the NLC mode only happened on a few days in 2010. We use the nominal temperature profiles from version 8 of MIPAS Level 2 data retrieved by ESA (Dinelli et al., 2021). Version 8 data from the other modes are obtained through the Institute of Meteorology and Climate Research in cooperation with the Instituto de Astrofísica de Andalucía (IMK/IAA) retrieval algorithm (García-Comas et al., 2012; Kiefer et al., 2021). The typical total errors are 0.5-2 K below 70 km and 2-7 K above (for MA, UA, and NLC modes). The typical vertical resolutions in the comparison range of this study are 3-7 km.**"

*132 – You use the line parameters from HITRAN to calculate absorption/emission spectra. Also, please indicate here what version(s) of HITRAN you're using*

We used both HITRAN 2016 and 2020. The description can be found in Section 3.1, lines 153-158.

*142 – should be "coefficients" as they are different for the two bands*

Revised as suggested.

*142 and after – the "n" in "n[x]" is not necessary as the square brackets already (typically) indicate number density*

Thanks for the suggestion. They have been revised.

*140-150 – It seems odd that you're discussing this in terms of density of "emitting" O2 instead of excited O2 in the specific state. The math is fairly straight forward, and all you need to add to the equation is a branching ratio, e.g. the Franck-Condon factor for the A-band.*

Thanks for the suggestion. The main reason we wanted "emitting" rather than "excited" is that there is another emission channel from $b^1\Sigma_g^+$ state to the $a^1\Delta_g$ state (i.e., the Noxon band). Since this work only focus on emissions at 0.76 and 1.27 μm, we think it is more appropriate to use "emitting".

*151 and after – please use the more standard variables λ and v_bar (nu with a bar over it) to represent wavelength and wavenumber*

We replace $w$ with $\lambda$ for wavelength, but we prefer keeping wavenumber as $v$ to be consistent with HITRAN (https://hitran.org/docs/definitions-and-units/).

*Figure 1 – could use a dashed vertical line in the middle to indicate the center of the line-of-sight/location of tangent height. Also, the description indicates that the tangent height is in the*

*middle of a layer, whereas here it looks like it is at the bottom of a layer. Please make it consistent.*

The tangent height is at the bottom of a layer. We clarify this by modifying lines 137-139 to

**"An atmospheric layer bounded by two tangent heights of SCIAMACHY limb observations is the basic spatial resolving unit of this study. We create an additional layer above the outermost tangent height by assuming a layer thickness equal to the average difference between adjacent tangent heights."**

A dashed vertical line is added to the figure as suggested.

*Section 3.3 – In optimal estimation, the measurement vector is typically represented by y (the retrieval function is typically R), so it's a bit odd having the measurement vector as r*

We see that point, but since the measurement vector elements are radiances, we would rather keep the same symbol ($r$) and just use bold symbol to indicate it is a vector.

*238 – I would suggest using something like "retrieval system" instead of "forward model" in order to be more encompassing*

Revised as suggested.

*250 – I would assume that results of the linear inversion would be prone to large, unrealistic oscillations that could lead to convergence issues. Is that the case? And if so, could some type of heavy regularization (smoothing) be applied to the result to get the profiles closer to a realistic estimate?*

Regularization will likely improve the linear inversion but it seems unnecessary. We did not use the full profile from linear inversion but rather take the mean value of this profile as a uniform a priori. The a priori error is 100 times of this value to guarantee the posterior is dominated by observation. To clarify, lines 251-252 are revised to

**"The corresponding prior error is set to be 100 times the prior, a constant value for all altitudes. This effectively gives no prior constraint to the [$O_2$*] profile and assures its information all comes from observations through near-unity DOFS of retrieved [$O_2$*]."**

*258 – was it mentioned earlier that this is performed in log space? If not, please explicitly state this prior to here and discuss the trade off of retrieving log values.*

Only the ground-state $O_2$ number density is retrieved in log space. This is because it's more natural to formulate the error of prior $O_2$ profile (from MSIS) as a percentage, instead of absolute $O_2$ number density. That sentence is updated to

**"In the state vector, we also include changes of the $O_2$ profile relative to the *a priori* from MSIS, which is equivalent to retrieving the natural log of [$O_2$]."**

*265 – why does the xi+1 variable have a "d" in front of it?*

Because this is the update of the state vector for the iteration.

*280 – The "airglow retrieval" is not attempted*

Revised as suggested.

*338 – What is meant by "over and above the mesopause region"?*

Revised to "**The emitting $O_2$ number density ($[O_2(b^1\Sigma_g^+)]$) can be retrieved with high precision above ~90 km.**"

*343 – I get what you're saying about the profile being "W" shaped, but it's not exactly intuitive what that means. If you want to describe it in that way, I'd say it's more "ε" shaped, but I'd suggest simply describing it as an inversion layer or a possible double mesopause.*

This sentence is revised to "**Both SCIAMACHY-based temperature profiles capture the double mesopause feature observed by ACE-FTS and agree well between each other at 80--95 km, where both airglow are strong and less interfered.**"

*382-383 – The secondary ozone peak has been studied for multiple decades now, so the Li et al. 2020 reference is not appropriate on its own.*

Thanks for the suggestion. We add a reference to Smith and Marsh, (2005, doi:10.1029/2005JD006298).

*393 – a value of 0.5 seems relatively low, as I usually expect ~0.7-0.8 as a cut-off point. Can you please discuss the distribution of "DOFS" values (what I would call "response" values) for the retrievals (e.g., what percentage of retrievals are rejected if you use 0.5, 0.75, etc., like what is done later in section 5)*

See the previous response about the DOFS. These are diagonal values of the averaging kernel, i.e., $\frac{\partial \hat{x}_i}{\partial x_i}$, not sum of rows. As shown by Figure 3c, the DOFS for temperature is a strong function of altitude and close to unity above ~50 km. It drops rapidly in the stratosphere due to stronger a priori constraint and diminishing airglow signal. Increasing the DOFS threshold will just slightly move up the lowest altitude of temperature profiles. The following two figures compare thresholds of 0.5 and 0.8 side-by-side:

[Figure]

We will keep the threshold as 0.5 as it shows the stratopause better. This sentence is modified to

**"Only temperature retrievals with DOFS greater than 0.5 are included in the averaging, and only grid cells with more than 5 measurements averaged are shown, which limits the temperature profiles to above ~40 km."**

*409 – Please explain what you mean by "due to horizontal heterogeneity at large SZA". Are you saying at larger SZAs there is more diurnal variation along the line-of-sight? If so, wouldn't it be for SZAs closer to 90° (i.e. sunset or sunrise), not necessarily larger?*

Yes, at these altitudes the sunset and sunrise happen at SZA slightly larger than 90°. We remove the ascending portion of an orbit and SZA above 100° when averaging individual orbits into the climatology. The sentence at line 409 is revised to

**"We believe those are retrieval artifacts potentially due to horizontal gradient of airglow intensity at sunrise or sunset conditions."**

*Figure 13 – please include the 1:1 lines*

The 1:1 lines are already there as the red dashed lines.

*436 – What is meant by "very consistent"? because that's not how I would necessarily describe those results.*

This sentence is revised to "**Overall, the correlation coefficients of mesopause temperatures from the two bands ranges from 0.75 to 0.90, with the *A* band temperature colder than the $^1\Delta$ band temperature by 5-8 K.**"

*440 – coincidence criteria*

Revised as suggested.

*Figure 14 – a color bar is needed to indicate altitude.*

Revised as suggested.

*478 – please quantify the findings of previous MIPAS temperature comparisons*

This sentence is revised to "**Garcia-Comas et al. (2014) compared MIPAS with a range of satellite and ground-based temperature observations and found that MIPAS temperature differs from others by 2 K at 50-80 km in spring, autumn and winter at all latitudes, and summer at low to mid-latitudes. Differences between MIPAS and the other instruments in the summer high latitudes are typically smaller than 2 K at 50-65 km and 5 K at 65-80 km. MIPAS in general shows colder temperatures in the mid-mesosphere.**"

*Figure 16 – Please use either a dashed line or maybe shading to indicate areas where the MIPAS data is from climatology*

We change the vertical ranges to only show the recommended vertical range of MIPAS nominal data (below 70 km). The updated Figure 16 is included here:

[Figure]

*500 – It's unclear what is meant by "due to horizontal heterogeneity of airglow"*

It is revised to

"… **strong warm bias at winter polar stratopause region likely due to horizontal gradient of airglow intensity that violates the homogeneous layer assumption for the retrieval algorithm.**"

*Math is not my thing, so I did not check the equations in the appendices. In my opinion, the appendices aren't necessary, but I'm not opposed to them.*

We hope to keep Appendices A and B as a formal documentation of the forward model and jacobians. The derivation of the emitting layer optical depth was an essential step, as otherwise the layers with strong self-absorption would be subject to unacceptable temperature biases. Appendix C is removed.